# Path length and sediment transport estimation from DEMs of Difference: a signal processing approach

Lindsay Capito[1], Enrico Pandrin[2], Walter Bertoldi[2], Nicola Surian[1], Simone Bizzi[1]

[1] Department of Geosciences, University of Padova, Padova, 35131, Italy

[2] Department of Civil, Environmental, and Mechanical Engineering, University of Trento, Trento, 38122, Italy

*Correspondence to*: Lindsay Capito (lindsaymarie.capito@studenti.unipd.it)

**Abstract.** The difficulties of measuring bedload transport in gravel bed rivers have given rise to the morphological method wherein sediment transport can be inferred from changes in riverbed elevation and estimates of the distance traveled by sediment, its path length. Because current methods for estimating path length are time and labor intensive, we present a method to estimate a characteristic path length from repeat digital elevation models (DEMs of difference i.e., DoDs). We propose an automated method to extract the spacing between erosional and depositional sites on the DoD by the application of Variational Mode Decomposition (VMD), a signal processing method, to quantify the spacing as a proxy for path length. We developed this method using flume experiments where bed topography and sediment flux were measured and then applied it to published field data with physical path length measured from tracer measurements. Our sediment transport estimates were not significantly different than the measured sediment flux at lower discharges in the lab. However, we observed an underestimation of sediment flux at the higher discharges in the flume study. We interpret this as a limit of the method in confined settings, where sediment transport becomes decoupled from morphological changes. We also explore how the time between survey acquisitions, the morphological active width relative to the channel width, and DoD thresholding techniques affect the proposed method and the potential issues they pose to the morphological method in general.

## 1 Introduction

In gravel bed rivers sediment transport fundamentally controls morphological processes but is notoriously difficult to measure due to its spatial and temporal heterogeneity (Hoey, 1992; McLean and Church, 1999), measurement uncertainty (Vericat et al., 2006), and the logistical challenges of field measurements. The morphological approach is a method to estimate bedload transport based on observed changes in morphology. There have been many implementations of the morphological method since its inception and it has been reviewed extensively (Ashmore and Church, 1998; Brewer and Passmore, 2002; Church, 2006; Vericat et al., 2017). With the increased availability of hydrologic data and modeling capabilities the morphological method has also been applied in two dimensions (x,y) by coupling a 2D hydraulic model to account for sediment routing (Lane et al., 1995; Antoniazza et al., 2019; Bakker et al., 2019). These 2D applications shed light on the functional links between topographic changes and spatial distribution of bedload transport. Antoniazza et al., (2019) quantified the potential errors in estimating sediment transport using a 1D approach where 2D cross-stream sediment fluxes are neglected. The error associated with neglecting the 2D fluxes may be especially important in multithreaded channels. They also explored how DEM accuracy and the frequency of acquisitions affect the estimates of sediment fluxes derived by the morphological method. These 2D applications enhance the accuracy of the morphological method to estimate sediment transport, however, these studies benefited from intensive field campaigns and an accurate accounting of upstream water and sediment supplies, often not available in real case studies. In this paper, the desire is to explore novel approaches to apply the morphological method using topographic data alone, as hydraulic and sediment supply data are not available in many applications and management situations.


The morphological method can be formalized based on the sediment continuity equation:
$$(Q_{b_{in}} - Q_{b_{out}})\Delta t = (1 - p)\Delta V , \tag{1}$$

where $Q_{b_{in}}$ and $Q_{b_{out}}$ are the volumetric sediment flux in and out of the reach respectively, $\Delta t$ is the time between
surveys, $p$ is the sediment porosity, and $\Delta V$ is the change in volume (Ashmore and Church, 1998; Church, 2006). The
sediment continuity equation can be solved in several ways, but in addition to $\Delta V$ measured from the DoDs, it requires
that either the incoming flux $Q_{b_{in}}$ or the outgoing flux $Q_{b_{out}}$ be defined. In most cases, neither of these fluxes are
known, as they are the exact parameters that need to be estimated when applying the morphological method. This
conundrum has been addressed by setting a zero-flux boundary, such as a dam or gravel sand transition (McLean and
Church, 1999), by segmenting the reach such that a zero-flux boundary is set between a section of net deposition to one
of net erosion (Vericat et al., 2017; Calle et al., 2020) or by measuring flux either into or out of the reach (Grams et al.,
2013; Antoniazza et al., 2019).
Alternatively, Eq. (1) can be modified so that active layer depth, $d_s$ and width $w_s$ ,and the virtual velocity, $v_b$ are used:
$$Q_b = v_b \, d_s \, w_s \, (1 - p)\rho_s \tag{2}$$


Where $w_s$ is the active layer thickness, generally measured by chains and estimated by depth of scour (Church and
Haschenburger, 2017), and $v_b$ is equal to $L/T$, $L$ being the distance the particles travel and $T$ the time over which the
particles are traveling (Church, 2006). The virtual velocity approach has been successfully applied using tracer gravels to
estimate the path length parameter $L$ in a variety of morphological settings (Liébault et al., 2012; Mao et al., 2017; Brenna
et al., 2019, 2020; Brenna and Surian, 2023). Unfortunately, tracer studies are time and labor intensive, requiring multiple
site visits and intensive recovery campaigns which often have low recovery rates, especially for painted clasts (Hassan
and Bradley, 2017; Brenna et al., 2019). Furthermore, tracer studies are often applicable only to exposed bars, ignoring a
large portion of in-channel transport, and can be sensitive to the seeding location (Liébault et al., 2012). To overcome
these limitations, several methods have been proposed to estimate path length based on the connection to morphology.
The term path length describes the distance traveled by a particle from entrainment to deposition during a transport event
and is punctuated by shorter bursts of movement termed step lengths (Einstein, 1937). Individual particles do not all
entrain, travel, and deposit together in unison but rather form a distribution of path lengths potentially dependent on grain
size, flow strength and duration, and channel morphology. The relative strength of these physical controls on path length
has been explored with varied results. Some studies have found relationships between path length and flow metrics such
as stream power (Hassan et al., 1992; Schneider et al., 2014; Vázquez-Tarrío and Batalla, 2019; Vázquez-Tarrío et al.,
2019) but a considerable scatter in the data has reinvigorated the debate over the role of morphology as a primary control
of path length (Hassan and Bradley, 2017; Vázquez-Tarrío and Batalla, 2019; Vázquez-Tarrío et al., 2019).
The connection between morphology and path length has long been discussed. Neill (1971) proposed that path length in
meandering rivers should be equal to the distance from an erosional site (eroding bank) to the next depositional site (point
bar) downstream. Many others have observed similar relationships based on the spacing of erosional and depositional
sites and channel morphology (Beechie, 2001; Pyrce and Ashmore, 2003a, b; Hundey and Ashmore, 2009; Kasprak et
al., 2015; Vázquez-Tarrío et al., 2019). Further, depositional areas (typically bars), have demonstrated a higher probability
of 'trapping' particles than erosional morphological units (McDowell and Hassan, 2020; McDowell et al., 2021). Finally,
experimental research has confirmed the preferential deposition of particles specifically at bar heads and margins even in
channels with more complex morphology, for example, in braided rivers (Kasprak et al., 2015) but it is reasonable to
assume that in multithreaded channels, multiple path lengths might exist at different flow stages in primary and secondary
channels.
The path length used for the virtual velocity approach is generally taken as the mean travel distance (Wilcock, 1997;
Vericat et al., 2017; Mao et al., 2017; Brenna et al., 2019). However, as we have seen from tracer studies, particles tend
to form a distribution of travel distances, therefore, it is unclear if the mean path length is the best representation of a
'characteristic' path length to estimate bedload transport. To obtain an estimate of reach scale sediment transport we want
to approximate the distance travelled by the bedload that builds geomorphic units, this is what we will consider as a
'characteristic path length'. Tracer studies have allowed us to see that this may not necessarily be the average distance,
as evidenced by the wide variety of path length distributions, it is often the case that many or even most (the mode) of
path lengths are very short, thus skewing the average depending on the distribution. For example, Pyrce and Ashmore
published synthesis of tracer studies and demonstrated that at formative discharges, particle path length distributions often
exhibit primary or secondary modes corresponding to the location of bars, where deposition occurs (Pyrce and Ashmore,
2003a). Further, flume experiments with tracers showed that the majority of particles eroded from an upstream scour pool
were deposited at the point bar apex and corresponded to peaks in bi or multimodal path length distributions (Pyrce and
Ashmore, 2005). Therefore, the characteristic path length, i.e., the most representative and sound value to be used in
sediment transport estimations, might be better described by these primary or secondary modes in channels with bar
morphology at channel forming flows.
If a characteristic path length can be inferred from changes in morphology as previously discussed, advances in
topographic survey techniques to acquire detailed digital elevation models (DEMs) and facilitate change detection,
provide an opportunity to streamline the estimation of sediment transport. The high-resolution topography (HRT)
revolution (Vericat et al., 2017) has provided an abundance of high quality surveys and an increased frequency of
change detection based on the differencing of DEMs to create digital elevation models of difference (DoDs)
(Brasington et al., 2000; Lane et al., 2003). Vericat et al. (2017) proposed an equation to use the path length with the
volume of erosion derived directly from the DoD
$$Q_b = (v_b \ \sum V_e \ (1-p)\rho_s)/L_c \, , \tag{3}$$
where $\sum V_e$ is the total volume of erosion from the DoD and $L_c$ is the length of the analyzed DEM by which the volume
of erosion is normalized (Vericat et al., 2017). To use this method, $L_c$ must be long enough for average path lengths (L)
to occur and $T$ must be short enough to prevent repeated erosion and deposition, known as compensation (Lindsay and
Ashmore, 2002).
Redolfi (2014) attempted to estimate the path length parameter directly from the DoD using the length of individual
erosional patches as a proxy for the length of the erosion-deposition sequence. This approach avoids the need to couple
each erosional area to a downstream depositional area, which can be difficult to automate in multi-thread rivers. While
this method scales well with flow metrics and provides reasonable estimates (Redolfi, 2014; Vericat et al., 2017), the

hypothesis that the length of erosional areas is equivalent to the erosion-deposition distance has not been tested in different morphologies, and it is not clear how the survey resolution may affect the estimates by fragmenting the erosional areas into smaller parts. Recently, Calle et al. (2020) used a method of river segmentation to visualize the pattern of erosion and deposition and infer sediment connectivity as well as to estimate potential travel distances. They defined boundaries between river segments and classified them into types based on their net erosional or depositional characteristics. Focusing on the "type 1 depositional boundary" wherein the upstream section is erosional and the immediate downstream boundary is depositional and depending on the volumes of deposition and erosion in these segments they were able to estimate minimum or maximum transport distances (Calle et al., 2020). This approach provides greater insight into the spatial connectivity of the river corridor and is useful to understand reach scale processes. However, depending on the river, and the sections surveyed, the number of type 1 boundaries may limit the applicability of the method in defining a characteristic path length and crucial information may be missed where the pattern of erosion and deposition is not clear, or the periodicity spans multiple sections. For example, where there are back-to-back patches of erosion or deposition or the overall pattern is separated by small areas of mixed boundaries.

Given the observations linking path length to morphology and building on the aforementioned methods, we seek to expand on the idea that characteristic path length can be inferred from changes in morphology at near transport event scale comparisons. If during a flood, sediment is mobilized from an area of erosion to an area of deposition as represented on the DoD, the distance between the two should correspond to a characteristic path length. Following these assumptions this work has the following objectives: i) to propose an objective and semiautomatic method to quantify a characteristic path length as represented by the periodic nature of erosion and deposition from the DoD using flume data; ii) to compare these estimates of a characteristic path length to measured path length distributions obtained from tracer data in the field; iii) and finally to evaluate the conditions in which a characteristic path length is appropriate to estimate sediment transport.

## 2 Methods

To meet our objectives, we use flume experiments at varying discharges with direct measurement of output sediment flux and sets of repeat DEMs from which DoDs are created and used to identify patterns of erosion and deposition. We then develop a semiautomated method to extract these distances between erosion and deposition as a proxy for the characteristic path length and then compare our estimates of sediment flux calculated using the characteristic path length to measured sediment flux. Finally, we compare the characteristic path length estimates from a published case study to the physical path length distributions as measured by tracers in the field to see how the characteristic path length corresponds to path length distributions.

### 2.1 Path length

A key assumption inherent in our objectives is that sediment moves from an area of net erosion to an area of net deposition during the time period between DEM acquisitions and that this represents a characteristic path length. Ferguson and Ashworth (1992) proposed a similar method of matching specific erosional and depositional patches albeit without the assistance of a DoD. This method was then implemented in the Sunwapta River, Canada (Goff and

Ashmore, 1994) although the authors note the difficulty in finding perfectly matching patches and conclude that
erosional and depositional processes are likely more dispersed. Here we will implement this "manual method" as a
means of comparison for the automated method presented later. The most obvious method to quantify this distance
between erosional and depositional sites on the DoD is to measure the spacing manually using a GIS program however,
this requires many subjective evaluations. Firstly, we must decide where on the patches of erosion and deposition to
begin and end the measurements. Because patches of erosion and deposition are not symmetrical or of equal size, the
distance between the two depends on which area of the patch we choose to begin and end the measurements. For
consistency, we choose the center of the patch (Fig. 1) after Ashmore and Church (1998). Next, we must determine
which patch of erosion matches with which patch of deposition which is not always obvious, and as noted previously,
likely does not accurately represent the nature of bedload transport (Goff and Ashmore, 1994). Here we perform this
method solely for comparative purposes and therefore used our knowledge of morphological processes to make a best
estimate. For example, a patch of erosion on an outside bend likely corresponds to the deposition of the next point bar
downstream (Fig. 1). Although this method is capable of producing crude estimates of path length to overcome the
aforementioned biases (Ferguson and Ashworth, 1992; Goff and Ashmore, 1994; Ashmore and Church, 1998) we
propose a method to estimate a characteristic path length without relying on the matching of erosion and deposition but
rather to use the periodic nature of these processes. Additionally, we seek to create a method that is both objective and
semiautomated.

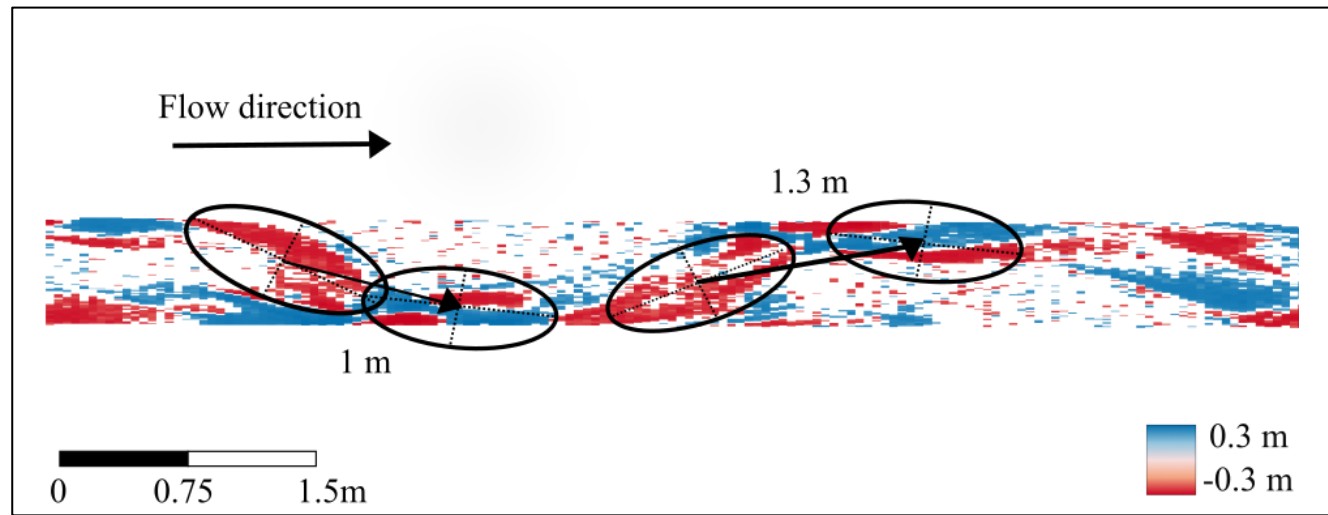

**Figure 1: Manual method to measure spacing of erosional patches (red) and depositional patches (blue) on a**
**DoD.**
**2.2 Semiautomated extraction of path length**
To visualize and then quantify the periodic nature of erosion and deposition from the DoD we simplify the spatial
heterogeneity of the DoD into a vector of the net change in elevation in a streamwise direction (Fig. 2a). Because
natural rivers are rarely straight, for field case studies, we must enforce a linear downstream directionality essentially
straightening the bends in the river. This is achieved by segmenting the DoD into a series of equally sized "bins" using
the segmentation tool of the Fluvial Corridor Toolbox (Roux et al., 2015) (Fig. 2a). The bin size can affect the pattern of
erosion and deposition in that by selecting too large of bins we may miss important erosional or depositional areas when
they are summed in the same bin. Similar methods that require river segmentation have proposed using the reach
averaged width for the length of the bins (McDowell et al., 2021) or half of the width of the reach (McDowell and
Hassan, 2020) although these studies had different objectives. Calle et al. (2020) applied a segmentation method with a
similar goal of identifying corresponding zones of erosion and deposition and set the bin sized based an assessment of
the river dimensions as well as the minimum transfer distance of interest. Therefore, depending on the river, the user
may select differently sized bins. Once the river is segmented, we then sum the values in each bin to obtain a vector of
the net change in elevation in a downstream direction (Fig. 2b). In the flume studies, where there is no sinuosity, we
simply sum each cross section of the DoD matrix. Oftentimes a reach is aggrading or incising and therefore the net
vector will have an increasing or decreasing trend (Fig. 2b). Because we are interested in the spacing between areas of
erosion and deposition rather than the overall trend, we remove it by subtracting a best-fit linear trend from the net
vector (Fig. 2b). Because we simplify the heterogeneity of erosion and deposition into a net vector of elevation change,
we risk compensating erosion and deposition within the same cross section, therefore we also create a vector of just
erosion and one of just deposition as well as the net allowing for a visual comparison of the relative contribution of
erosion and deposition to the net as well as the periodicity of the individual processes (Fig. A1). We can see that there
appears to be a periodicity as the net vector oscillates forming peaks and troughs and although this periodicity seems
apparent, quantifying the distance is not straightforward.

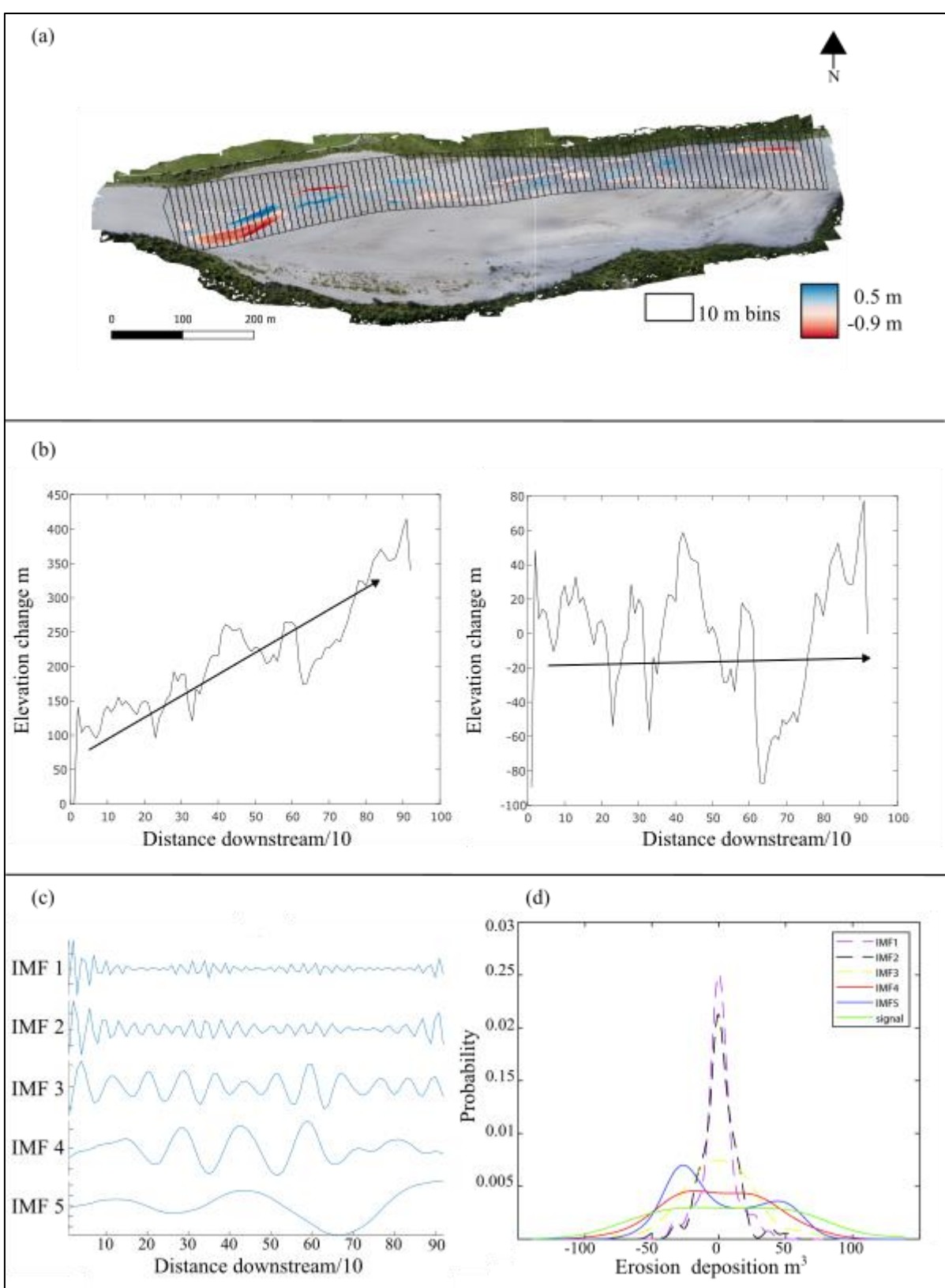

**Figure 2: VMD- HD method (a) Segmentation of the DoD (example orthophoto and DoD from the Tagliamento River, Italy). (b) Plot of the net original and detrended vector. (c) Variational mode decomposition (VMD) with 5 intrinsic mode functions (IMFs). (d) Probability density function (PDF) of each IMF and the original net vector.**

One approach could be to count the zero crossings and then use that distance as the proxy for path length. However, we risk measuring low magnitude spikes that cross zero that may not necessarily represent the overall periodicity or large oscillations that don't cross the zero line. A smoothing filter may be used to remove these low magnitude oscillations

but we risk losing potentially relevant information. To solve this problem, we turn to the realm of signal processing
where the practice of "denoising" and extracting information from oscillations is ubiquitous.
Signal processing is a field that deals regularly with extracting information and patterns that are not visually apparent
and its applications have been used in a wide variety of settings including voice recognition (Sigmund, 2003; Upadhyay
and Pachori, 2015), medical applications (Boudraa et al., 2005; Liu et al., 2008), and even time series analysis of
climate data (Barnhart and Eichinger, 2011). There are many approaches to de-noising including Fast Fourier transform,
empirical mode decomposition (EMD), and wavelet analysis. Each of these methods come with inherent strengths and
weaknesses, for example wavelet analysis requires that a mother wavelet be selected a priori and may influence the
results (Boudraa et al., 2005). We chose to use variational mode decomposition (VMD) due to its robustness with
respect to sampling and noise and the ability to handle signals that exhibit non-linearity and non-stationarity
(Dragomiretskiy and Zosso, 2014; Huang et al., 2016; Ma et al., 2017). VMD decomposes the signal into a set of
intrinsic mode functions (IMFs) each with a different central frequency (Dragomiretskiy and Zosso, 2014; Ma et al.,
2017) (Fig. 2c). In this case of our static 'signal' the frequency is more accurately described as the wavelength. It is
beyond the scope of this paper to describe the mathematics of VMD in detail, therefore, for a complete explanation see
(Dragomiretskiy and Zosso, 2014; Huang et al., 2016; Ma et al., 2017; Upadhyay and Pachori, 2015).
Once the original net vector of erosion and deposition is decomposed into the various IMFs, we need to select the IMF
or IMFs that most accurately represent the periodicity of the original data and therefore our characteristic path length.
Ma et al. (2017) proposed a method to select the most relevant IMF, and therefore periodicity of the signal, by
computing the probability density function (PDF) using kernel density smoothing for each of the five IMFs and of the
original data vector (Fig. 2d), then to calculate the Hausdorff distance (HD), a metric of geometric similarity, between
each IMF's PDF and the PDF of the original data and select the IMF most geometrically similar to the original data (Ma
et al., 2017) (hereafter VMD-HD method). In most cases, the longer wavelength IMFs most closely resemble the
original signal whereas the IMFs with shorter wavelengths are more likely associated with noise (Boudraa et al., 2005).
The computed wavelength is converted to a meaningful physical quantity by multiplying by the bin spacing in meters.
Because we are interested in the distance from peak to trough, we divide the period by two to obtain the path length
proxy (Neill, 1971; Ashmore and Church, 1998). Although this method allows for the selection of one IMF to
presumably represent the periodicity of the data, we record path lengths calculated from the other IMFs to evaluate the
range of estimates generated by the decomposition and determine if the VMD-HD method is appropriate for
determining a characteristic path length and the relative importance of other IMFs. All calculations were performed in
MatLabR2020b using the built in VMD function and the Hausdorff distance function (Danziger, 2023).
One important consideration when using VMD to decompose a signal is that is the user must define the number of IMFs
beforehand. The number of IMFs is important as under binning, choosing too few IMFs, may mean that critical IMFs
are missed, whereas over binning, can cause duplication of components (Wu et al., 2020). In signal processing, there are
sophisticated methods for determining the number of IMFs, for a summary see (Wu et al., 2020). However, for our
purposes and simplicity's sake, we performed a brief sensitivity analysis based on the property of convergence often
used in the signal processing methods (Wu et al., 2020; Huang et al., 2016; Ma et al., 2017). The default setting in the
MatLab function is 5 IMFs, we used 3, 5, 8, 15, and 25 IMFs to calculate path length and assessed how it changed for
the maximum IMF (Fig. A2). We found that using more IMFs generally increased the number of high frequency
components rather than the lower frequency IMFs (Fig. A3). Because these higher frequencies are generally associated
with noise and in our case are physically too small to likely represent meaningful path lengths (on the order of
millimeters) we decided more than 5 IMFs did not contribute physically meaningful information in that the IMFs with
longer wavelengths did not change drastically. We also determined that 3 IMFs were too few as it was clear that the
longer wavelengths were missing (Fig. A3). Therefore, we chose to use the default 5 IMFs as this provided a
manageable number of components while effectively separating the lower frequencies. This is a convenient starting
point for assessing the VMD method as a tool to extract the periodicity as a proxy for characteristic path length but is by
no means the only option. We encourage further exploration of the IMF parameter in future applications and as the
method is refined.
**3 Flume and field data**
The method was tested using data from a set of flume runs performed in the Hydraulic Laboratory of the University of
Trento, where DEMs were generated for fixed time intervals and varying discharges, and direct measurements of the
bedload flux were also collected. To test the efficacy of the method in the field, we selected a published dataset of
measured path lengths with corresponding DoDs for the San Juan River in British Columbia Canada (McQueen et al.,
2021). Although McQueen et al. deployed tracers in four separate periods, there was only one deployment (2018-2019)
with corresponding DEMs (McQueen et al., 2021). DoDs and corresponding tracer data were available for three
separate sites (bar 6, bar 7, and bar 15) for the 2018-2019 period. Detailed information on their collection and
processing can be found in McQueen et al., 2021.
**3.1 Flume experiments**
The Trento laboratory experiments were carried out in a 0.6 m wide and 24 m long flume, filled with nearly uniform 1
mm diameter sand. The flume slope was set to 0.01 m/m. Topographic surveys were performed over the final 14 m of
the flume, to limit the upstream inflow effects, using a laser gauge, mounted on a movable deck. The longitudinal and
crosswise spacings were 0.05 m and 0.005 m, respectively. Four sets of nine runs were performed, with the flow
discharge set to 0.7, 1, 1.5, and 2 l/s, which correspond to a range of different planform morphologies (Table 1).
Sediment input at the upstream end of the flume was constant in each run, with a flux equal to the average measured at
the downstream end, as computed in a preliminary set of experiments. Therefore, the overall average bed elevation of
the runs was in equilibrium, with no net erosion or deposition. The runs were performed following the same procedure,
involving three phases of different lengths, based on the transport condition of each discharge. These durations were
estimated referring to the time scale for morphological evolution computed from the sediment balance mass equation
(Garcia Lugo et al., 2015), which can be expressed as:
$$T\_ex = \frac{DW^2}{Q_b},$$
(4)

where D is the average flow depth and W is the flow width. Table 1 provides the values of $T\_ex$ for each flume
experiment.
**Table 1: Initial conditions for each dataset including the type of validation data.**

|  | Flume 1 | Flume 2 | Flume 3 | Flume 4 | San Juan Bar 6 | San Juan Bar 7 | San Juan Bar 15 |
|---|---|---|---|---|---|---|---|
| Peak discharge (m$^3$/s) | 0.0007 | 0.001 | 0.0015 | 0.002 | 942 | 942 | 942 |
| Slope (m/m) | 0.01 | 0.01 | 0.01 | 0.01 | 0.0038 | 0.0031 | 0.0009 |
| Width (m) | 0.6 | 0.6 | 0.6 | 0.6 | 150 | 150 | 130 |
| D$_{50}$ (m) | 0.001 | 0.001 | 0.001 | 0.001 | 0.05 | 0.056 | 0.042 |

| | | | | | | | |
|---|---|---|---|---|---|---|---|
| Time scale T_ex (min) (eq.4) | 94 | 50 | 38 | 30 | - | - | - |
| Time between surveys (min) | 47 | 25 | 19 | 15 | ~1 year | ~1 year | ~1 year |
| ω* Dimensionless stream power | 0.15 | 0.22 | 0.33 | 0.43 | 0.76 | 0.61 | 0.31 |
| Validation Data | Sediment Flux | Sediment Flux | Sediment Flux | Sediment Flux | RFID tracers | RFID tracers | RFID tracers |
| Planform | Wandering | Wandering | Wandering transitional | Alternate bar | Wandering | Wandering | Wandering |


First, an initial phase of about 12 times this time scale $T\_ex$ with constant flow was run, to ensure the formation of a
near-equilibrium morphological condition, starting from a flat sand bed scraped to the prescribed slope. This was
followed by a long run, at constant discharge, lasting 19 times the time scale $T\_ex$, aimed at measuring the output
sediment flux. This was continuously monitored at the channel outlet, through a permeable basket placed on four load
cells. Sediment flux was measured every minute. After a bed topography survey, the third phase was a sequence of nine
shorter runs, lasting 0.5 times the time scale $T\_ex$, each followed by a bed topography survey, which produced nine
corresponding DoDs. The duration of these nine runs (and therefore the time interval between surveys) was decided to
have easily measurable changes of the bed morphology, without having significant compensation processes.
The DoDs were created by subtracting two consecutive DEMs, then underwent a three-step filtering process to highlight
the relevant erosion and deposition patterns, removing most of the noise associated with the surface roughness and
measurement accuracy. First, the DoDs were filtered considering a uniform detection threshold equal to 2 mm (2 times
the $D_{50}$), meaning that erosion or deposition values lower than this threshold are set to zero. Thereafter, a spatial average
was performed as a moving average on three values along the transversal direction where the DoD discretization is the
finest. Lastly, a despeckling algorithm removed all isolated cells, both considering single cells that show erosion or
deposition, as well as single cells that show no change. This last step was implemented to keep the detection threshold
as low as possible while removing unphysically small areas. Additionally, we calculated the morphological active width
by determining the percentage of the DoD that showed morphological activity (i.e., was not zero after filtering).
**3.2 San Juan River data**
To compare the characteristic path length to measured path length distributions in the field, we used data from the San
Juan River, located on Vancouver Island, British Columbia with a drainage area of approximately 730 km$^2$ and a mainly
rainfall driven hydrology (McQueen et al., 2021). The reach of interest in this study was alluvial in nature with a
wandering morphology and a substrate composed of gravel, cobble, and sand (McQueen et al., 2021). The time in
between acquisitions is one year, in which it is estimated there were five flood events able to generate sediment
transport using a threshold of 500 m3 s−1, which was visually estimated by the authors to be equivalent to the bankfull
discharge (McQueen et al., 2021). DEMs were generated by LiDAR acquisitions and have a spatial resolution of 10 cm
and a vertical root mean square error lower than 10 cm. Topographic changes between survey dates were then
calculated by processing the LiDAR DEMs using the Geomorphic Change Detection (GCD) software (Wheaton et al.,
2010). More information on how they were obtained and processed including the spatially variable thresholding
techniques can be found in McQueen et al. (2021). The LiDAR-derived DoDs were used to interpret patterns of tracer
displacement and burial depths and to provide information on the morphological development of the bars during the
study period. However, they do not provide complete reach-scale sediment budgets due to the lack of in-channel
topographic data and stage differences during each LiDAR survey affecting the relative portion of the river bed that was

exposed. The submerged area represented 22% of the DoD for bar 6, 42% for bar 7, and 36% for bar 15. Nevertheless, we believe the exposed part of the channel, the bars, and associated patches of erosion and deposition (see Fig. 9b) are sufficient to be used with our proposed method to estimate path lengths and be compared with field measured path lengths from the tracer data as a first application to field data. This is because we are not calculating sediment flux for the San Juan River and are only interested in comparing our estimates of the characteristic path length to the measured tracer distributions. As far as the pattern of erosion and deposition and how that may be disrupted, we recognize that the pattern could change by including the underwater areas, however, looking at figures 15 and 16 from McQueen et al. (2021) we can see that the tracers were largely recovered from the exposed bar surfaces in the 2018-2019 deployment. This gives us confidence that the deposition we are measuring corresponds largely to the deposition associated with the tracers. Although this is not an ideal situation, we believe the benefits outweigh the limitations considering the difficulty of finding high quality RFID tracer data and corresponding DoDs. The San Juan River DoDs were downloaded directly from the Scholars Portal Dataverse (https://doi.org/10.5683/SP2/UQGZCG). The DoDs were segmented using similar principles to Calle et al. (2020) in a similarly sized river, therefore the bin size was conservatively set at 10 m.

### 3.3 Validation and error estimation

Each study had unique initial conditions including slope, discharge, grain size, channel configurations, and time/flood events between surveys (Table 1). Because the studies vary with respect to these initial conditions, we calculated the dimensionless stream power ($\omega^*$) after Bertoldi et al. (2009) to compare them as:

$$\omega^* = \frac{Q \cdot S}{W \sqrt{g \Delta D_{50}^3}}, \tag{5}$$

where $Q$ is the peak discharge, $S$ is slope, $W$ is the average wetted width, $\Delta$ is the relative submerged density, $D_{50}$ is the median grain size, and $g$ is the acceleration due to gravity.

For the flumes, we used estimates of path length generated by the VMD-HD method and those associated with the two longest wavelengths, IMF 4 and IMF 5 separately to calculate the virtual velocity Eq. (2) and sediment flux Eq. (3) which we then compared to measured flux data. The measured sediment flux during the initial long run showed high variability, with phases of high and low sediment flux lasting several tens of minutes. For this reason, we prefer to use the data from the long runs, from which we estimated an average sediment flux of 0.33 g/s (SD=0.17) for the 0.7 l/s discharge, 0.78 g/s (SD=0.31) for the 1 l/s discharge, 1.98 g/s (SD=0.65) for the 1.5 l/s discharge, and 3.22 g/s (SD=0.79) for the 2 l/s discharge. We subdivided the second phase into 38 intervals of 0.5 $T\_ex$ duration, equal to the duration as the short runs in phase 3, and computed the variability of the flux over this range.

We used ANOVA to compare path length, virtual velocity, and erosion across the four discharges ($\alpha$=0.05) and a Post-hoc Tukey test to explore significant differences between discharges. To compare the measured sediment flux to the estimates from the VMD-HD method and the IMF 4 and IMF 5 estimates we used a student's t-test ($\alpha$=0.05). And finally, to compare the error of our path length and sediment transport estimates we calculated the relative percent error $\delta$ in order to compare the sediment flux estimates to that of the long runs of average sediment flux as:

$$\delta = \frac{|E - M|}{M}, \tag{6}$$

where $E$ is the average of the estimated sediment flux for the 9 runs at a given discharge and $M$ is the averaged
measured sediment flux from the long run at the same given discharge. For the San Juan River we compared the VMD-
HD estimates of path length and IMFs 4 and 5 qualitatively to the published path length distributions and the locations
of mean, median, and modes. The tracer recovery locations were accessed in spreadsheet form and in keeping with the
analysis of the authors we disregarded any tracers that moved less than 10 m before calculating the path length
distributions.
**4 Results**
**4.1 Flume experiment**
To aid in the interpretation of the results, Fig. 3 shows a DoD from the lowest discharge, 0.7 l/s (a) and the highest
discharge, 2 l/s (b) with the net vector (continuous line), IMF 4 (dashed line), and IMF 5 (dotted line) as obtained from
the VMD method. Oftentimes the areas of deposition and erosion from the DoD correspond clearly to the IMF 4 and 5
vectors as with the 0.7 l/s discharge where areas of deposition are concave and areas of net erosion correspond to
convex areas of the vector (Fig. 3a). At the higher discharges (1.5 l/s and 2 l/s) the total area of morphological activity
increases and patches of erosion and deposition begin to overlap, creating a more chaotic and difficult to discern pattern
(Fig. 3b, A1). We also observed a similar periodicity in the erosional and depositional vectors and at the 2 l/s discharge
the depositional vector appears to show this most clearly (Fig. A1).

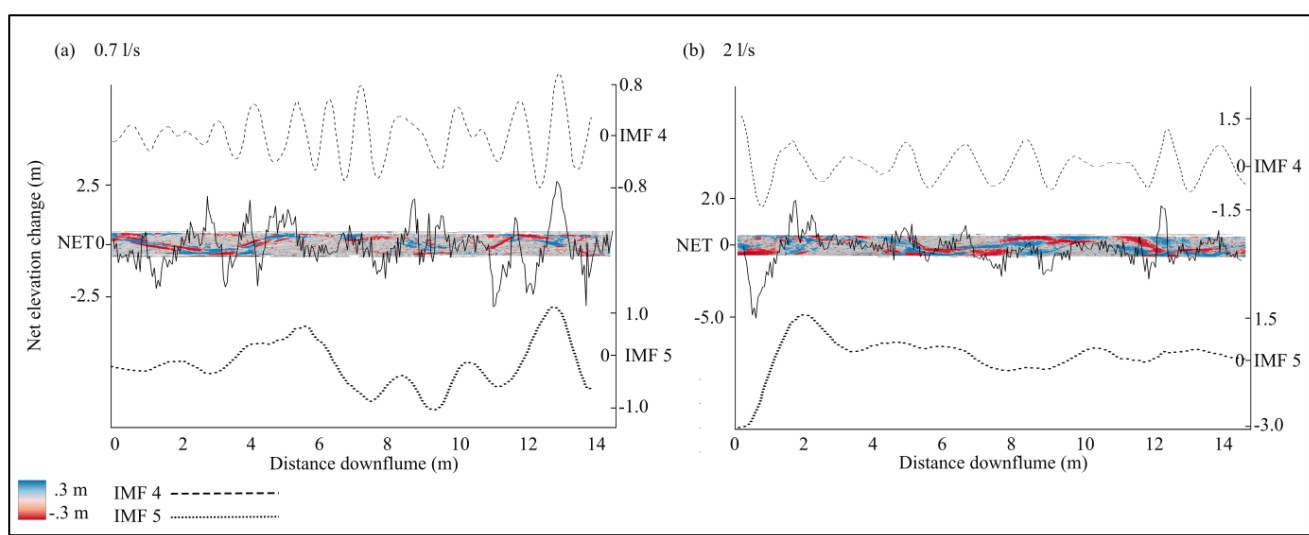


**Figure 3: DoDs from 0.7 l/s discharge (a) and 2 l/s discharge (b) with the net vector of elevation change laid over**
**the top. IMF 4 vector (above, dashed line) and IMF 5 vector (below, dotted line)**
In the flume experiment, the VMD-HD method of choosing the most relevant IMF selected the longest wavelength IMF
5 71% of the time and IMF 4 23% of the time. IMFs 2 and 3 were never selected and IMF 1 was selected only twice.
However, at the higher discharges (1.5 l/s and 2 l/s) IMF 4 was selected more frequently, thereby reducing the average
path length when compared to the lower discharges (Fig. A4). Using the selected IMFs, the VMD-HD method
estimated a similar average path length for all of the discharges (Fig. 4). The averages were, 1.45 m (standard deviation
(SD) = 0.93) for the 0.7 l/s discharge runs, 1.24 m (SD=0.58) for the 1 l/s runs, 1.21 m (SD = 0.58) for the 1.5 l/s runs,
and 1 m (SD = 0.37) for the 2 l/s runs (Fig. 4). The path length estimates derived from IMF 4 were similar for all
discharges, 0.51 m (SD=0.12) for the 0.7 l/s discharge, 0.55 m (SD=0.16) for the 1 l/s discharge, 0.56 m (SD=0.91) at
the 1.5 l/s discharge, and 0.46 m (SD=0.15) at 2 l/s (Fig.4) with no significant differences between the discharges (p >
0.05). The path lengths derived from IMF 5 were also similar between the discharges with no significant differences (p
> 0.05) and were, 1.75 m (SD=0.79) for the 0.7 l/s discharge, 1.55 m (SD=0.24) for the 1 l/s discharge, 1.79 m
(SD=0.67) for the 1.5 l/s discharge, and 1.37 m (SD=0.39) for the 2 l/s discharge (Fig. 4). The VMD-HD method
matched closely with the manually measured distances and there were no statistically significant differences for any of
the discharges (p-value > 0.05) (Fig. 4) while the IMF 4 and IMF 5 derived path lengths bracket the manually measured
distances and the VMD-HD selected path lengths (Fig. 4).

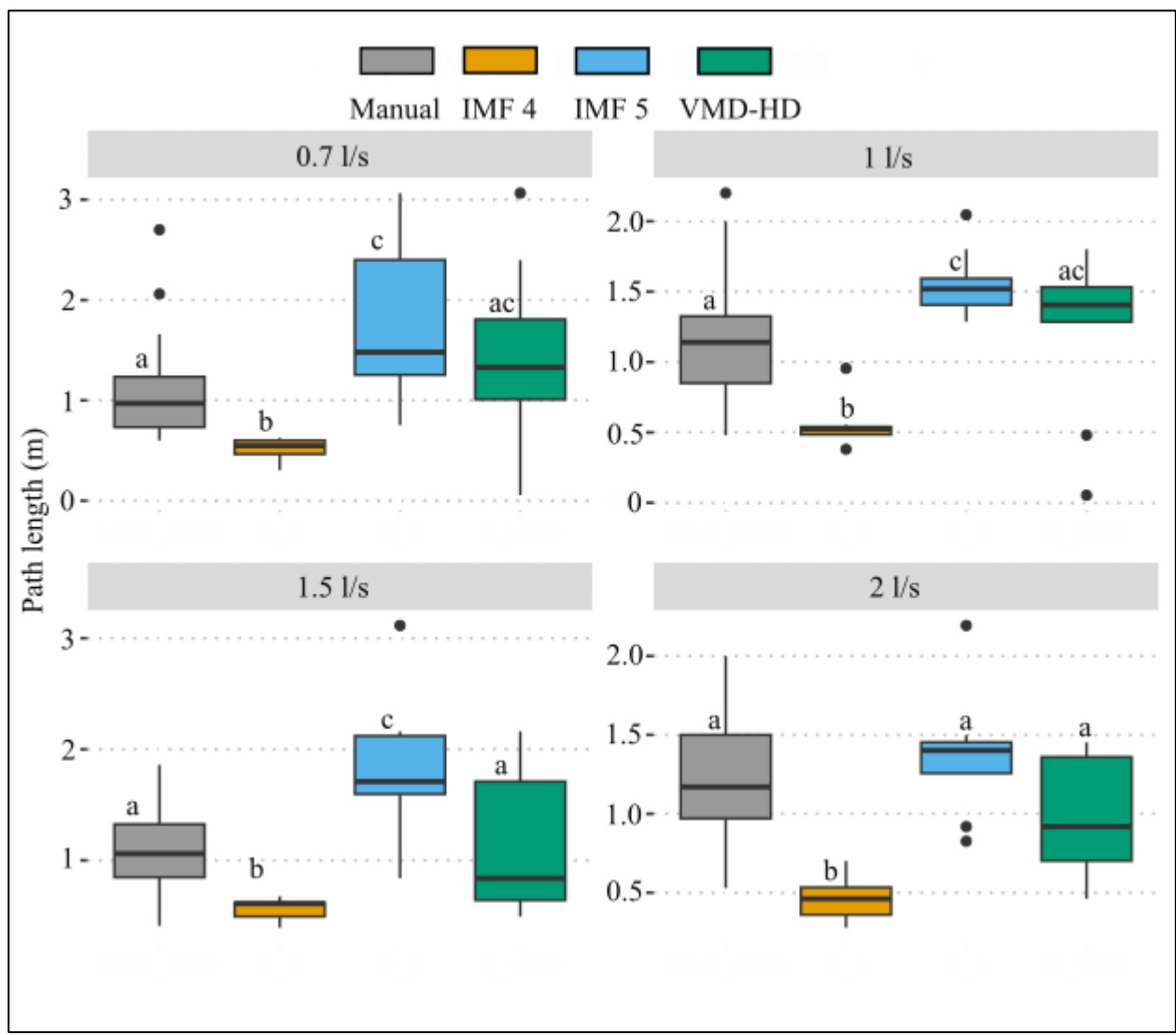


**Figure 4: Path length estimates from the manual method (gray), IMF 4 (orange), IMF 5 (blue), and the VMD-HD**
**method (green). Significant differences from the post-hoc Tukey test are denoted by letters a-c.**
The estimated path lengths were not significantly different between the discharges (p-value >0.05) and showed no
obvious trend of increasing or decreasing with discharge. However, when used to calculate the virtual velocity ($v_b$)
wherein the path length is divided by the time between surveys (Table 1), we see an increase in the virtual velocity with
discharge (p-value < 0.05) (Fig. 5). Likewise, the average volumes of erosion and deposition calculated from the filtered
DoDs increases significantly with discharge (p-value < 0.001) (Fig. 5).

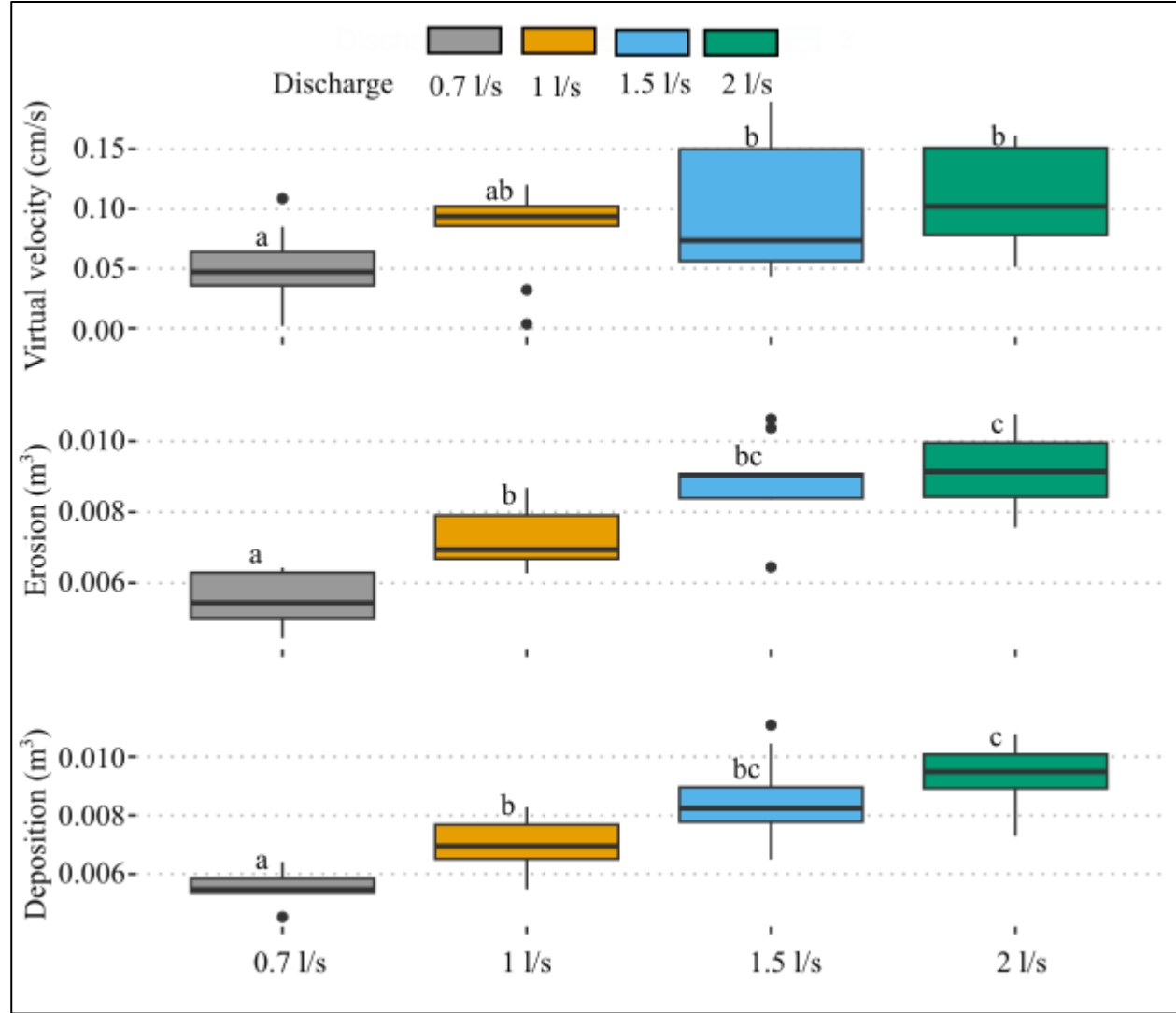

**Figure 5: Estimated virtual velocity using the VMD-HD path length estimates, measured volumes of erosion and deposition for each discharge. Significant differences from the Post hoc Tukey test are denoted by letters a-c.**

When used to calculate sediment transport Eq. (3) the VMD-HD method corresponds well to the measured average for the lower discharges (0.7 l/s and 1 l/s) whereas at the higher discharges (1.5 l/s and 2 l/s) the method significantly underestimated the measured flux (Fig. 6). For the 0.7 l/s discharge, the VMD-HD method estimated a rate of 0.39 g/s (SD = 0.25) averaged over the nine runs, which was not significantly different than the measured average of 0.33 g/s (SD = 0.18) and the relative percent error ($\delta$) was 18%. For the 1 l/s discharge the method estimated 0.81 g/s (SD = 0.38) and was not significantly different than the measured average of 0.78 g/s (SD= 0.30, $\delta=4\%$). At the higher discharge of 1.5 l/s the average estimated by the VMD-HD method was 1.33 g/s (SD = 0.82) whereas the measured average was 1.98 g/s (SD = 0.70) (p-value < 0.05, $\delta=32\%$). Finally, for the 2 l/s runs the estimated average was 1.41g/s (SD = 0.48) whereas the measured average was 3.22 g/s (SD = 0.98) (p-value < 0.001, $\delta= 56\%$) (Fig. 6).

If we use just the IMF with the longest wavelength (IMF 5) to estimate path length and calculate sediment transport, we slightly overestimate sediment transport at the 0.7 l/s discharge, 0.48 g/s although not significantly (p > 0.05, $\delta= 45\%$) (Fig. 6). At the 1 l/s discharge IMF 5 significantly overestimates the average flux with an estimate of 1.03 g/s (p < 0.01, $\delta= 32\%$). At the 1.5 l/s discharge the estimated flux of 1.88 g/s using the IMF 5 path lengths was not significantly

398    different from the measured flux (p > 0.05, δ= 5%) (Fig. 6). However, using the IMF 5 path lengths still significantly

399    underestimated sediment flux at the 2 l/s discharge, 1.95 g/s (p < 0.001, δ= 39%) (Fig. 6).

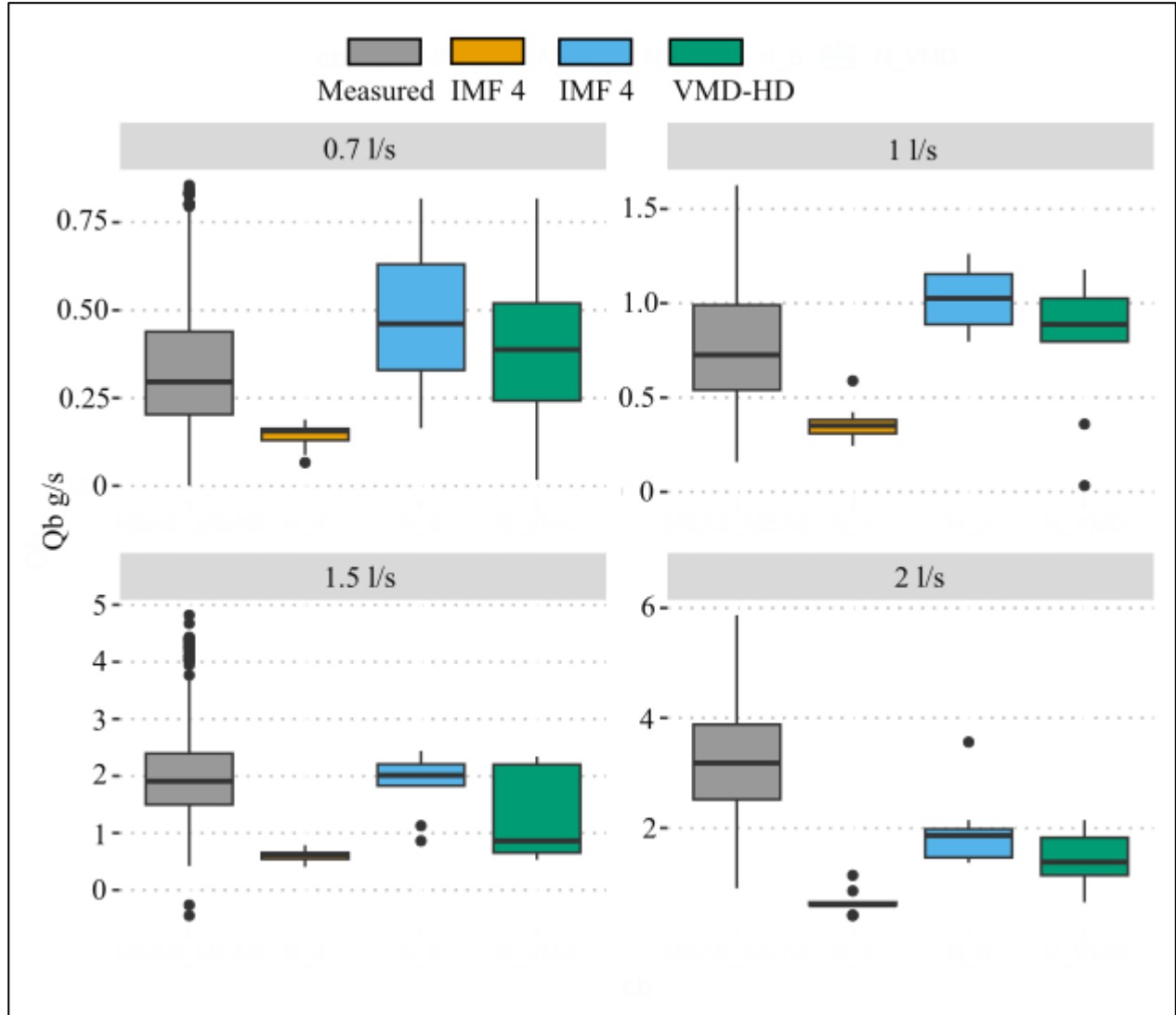

**Figure 6: Measured sediment flux (gray) compared to the estimates calculated using IMF 4 (orange), IMF 5 (blue), and the VMD-HD method (green). Significant differences from the post-hoc Tukey test are denoted by letters a-c.**

Using the second longest wavelength, IMF 4, we underestimate at all of the discharges (Fig. 6). The estimated flux was
0.14 g/s at the 0.7 l/s discharge (δ= 58%), 0.36 g/s at the 1 l/s discharge (δ= 54%), 0.61 g/s at the 1.5 l/s discharge
(δ= 69%), and 0.65 g/s at the 2 l/s discharge (δ= 80%), (all p values < 0.001) (Fig. 6).
**4.2 San Juan River**
The 2018-2019 year for which we conducted our analyses, was moderate in terms of excess flow energy with 5 flood
events exceeding a discharge of 500 $m^3 s^{-1}$ and a peak discharge of 942 $m^3 s^{-1}$. The path length distributions of bar 7
and bar 15 are positively skewed although there is a secondary mode in the bar 7 distribution corresponding roughly to
the bar tail (Fig. 7) whereas the distribution of bar 6 is bi-modal with the primary mode corresponding to the bar apex
(Fig.7). This is potentially because bar 6 had the most pronounced curvature, perhaps contributing to the clustering of
deposition just before the apex, where a migrating gravel sheet terminated (McQueen et al., 2021). This bar apex
corresponds with the path length from IMF 5 of 217 m which was selected from the VMD-HD method (Fig. 7). IMF 5
was also selected by the VMD-HD method for bar 7 equaling 324 m, and here we see a correspondence to the small
secondary mode where the authors note there was a clustering of tracers (Fig. 7) (McQueen et al., 2021). Again, IMF 5
with a path length of 323 m was also selected for bar 15 and corresponds closely to the bar apex, although there was not
a clustering of tracer deposition in this deployment as observed in the year with higher discharge (Fig. 7). Additionally,
bar 15 had the highest proportion of sand which is not represented by the tracers, potentially contributing to the
discrepancy between our estimates and the tracers. IMF 4 was always well below the lengths associated with the bar
apexes, and the median and mean tracer distances (Fig. 7). However, the bar apexes and the median and mean tracer
distances were always between IMF 4 and IMF 5 (Fig. 7). The range between IMF 4 and IMF 5 accounted for 62% of
the path length distribution for bar 6, 36% for bar 7, and 45% for bar 15.

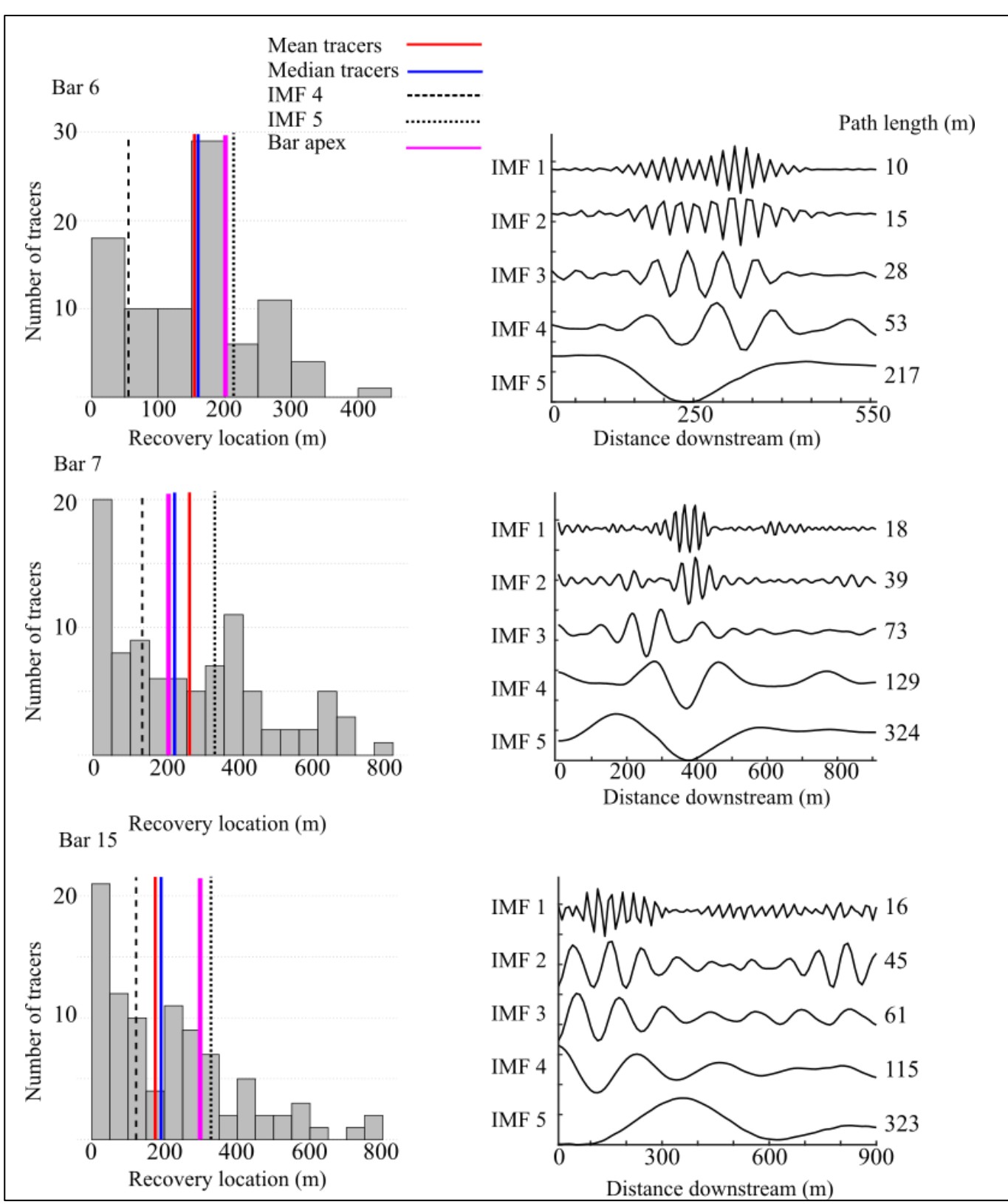

**Figure 7: Tracers-based path length distributions (on the left) and VMD derived IMFs for bars 6,7, and 15 from the San Juan River dataset. IMF 4 (dashed line), IMF 5 (dotted line), mean tracer distance (red), median tracer distance (blue), and the bar apex (pink) are shown over the path length distributions.**

**5 Discussion**

We developed a method to estimate the characteristic path length during a given flood using information inherent to the DoD by applying the principle that at channel-forming flows, the majority of particles move from an area of erosion to

the next area of deposition downstream (Pyrce and Ashmore, 2003a, b). By using the periodic nature of erosion and
deposition we overcome the subjectivity and time involved in measuring these distances manually while aligning
closely with these manually measured distances (Fig. 4). When evaluating the efficacy of our proposed method it is
important to keep in mind the uncertainty of even direct measurement of sediment transport. The spatial and temporal
frequency required to overcome the noise of measurement uncertainty (i.e., achieve an acceptable signal to noise ratio)
in some cases can require sub-daily monitoring with precise equipment (Grams et al., 2019). The variability of sediment
transport measurements in the flume study ranged from a standard deviation of approximately 30% to over 50% of the
averaged flux (Fig. 6). Given this high variability, our reach scale averages were not significantly different from the
measured averages for the 0.7 l/s and 1 l/s discharges (Fig. 6). Importantly, we observed that the method underestimates
the sediment flux significantly for the two highest discharges in the lab where the bed shows a higher percentage of the
width experiencing topographic change (Fig. 6). The method presented to estimate a characteristic path length using
only remotely sensed data shows promising results under certain conditions and provides insight into conditions where
it is not applicable.
**5.1 Path length estimation by VMD-HD method: limitations and perspectives**
**5.1.1 Flow effects**
Previous studies have shown a relationship between path length and hydrologic variables such as discharge, stream
power, and excess shear stress (Hassan et al., 1991; Pyrce and Ashmore, 2003b). A notable result of the flume
experiment is that the estimated path length did not significantly differ between the four discharges (Fig. 5). We
propose two possible explanations for this discrepancy with the literature. First, it is possible that the actual path length
is increasing with discharge as has been observed in previous studies (Hassan et al., 1991; Pyrce and Ashmore, 2003b)
but the method fails to capture it because the VMD-HD method is based on the spacing of erosion and deposition which
does not change for the varying discharges under the flume conditions. It is possible that at higher discharges the
characteristic path length, that we define as the distance from net erosional areas to net depositional areas, is not
appropriate under the higher flow conditions because most particles are moving farther than the next depositional site
downstream. This violates the assumption on which our method is based and is impossible to prove in our experiment
without tracers. We can however look to literature to understand the conditions in which tracers tend to travel more than
one morphological unit ( Liébault et al., 2012; Vázquez-Tarrío et al., 2019) and coupled with future studies, perhaps
determine the conditions under which a characteristic path length is inappropriate to estimate sediment transport.
From the San Juan River data we see that for a year with moderate flow, as characterized by the authors, that very few
tracers traveled further than the first depositional site downstream of their insertion point although it is possible that the
unrecovered tracers escaped the first bar, the recovery rates were high with 75% of tracers recovered for bars 6 and 15,
and 79% recovered for bar 7 (McQueen et al., 2021). However, the moderate flow year for which we had corresponding
tracer and DoD data resulted in two of the three sites with positively skewed distributions, and only bar six showing a
mode near the bar apex, which also corresponded to the IMF 5 path length (Fig. 7). The moderate flow conditions could
explain why our estimates lined up more closely with the bar apex for bar 15, where in the previous high flow year the
majority of tracers were deposited resulting in a symmetrical distribution (Fig. 8 from McQueen et al., 2021). It could
be that our method is strongly influenced by the morphology of the channel such that when flow is insufficient to create
symmetrical or bi modal path length distributions, we overestimate by using the characteristic path length because the
majority of particles are not reaching the next major depositional site downstream (i.e., a positively skewed
distribution). Additionally, when the flow exceeds a yet unidentified threshold, the majority of particles move more than

one depositional site downstream and therefore we underestimate sediment transport by using the characteristic path length. We can speculate that this is happening to some extent in the flume experiment. We see that at the lowest discharge, 0.7 l/s, we slightly overestimate the sediment flux especially using IMF 5 (Fig. 6) and underestimate the flux at the highest discharge, 2 l/s, where we also see a simplification of channel morphology (Fig. 6, 3). Because we did not have tracers in the flumes we can not say if the path length distributions were in fact different between the lowest and highest discharges. Future applications of this method with tracer data both in the flumes and in the field could help to understand when the characteristic length scale of morphology extracted by the method is an appropriate estimate of sediment transport and if this corresponds to flow metrics and path length distributions. In the flume studies we tested the idea that the majority of particles are bypassing the first depositional site simply by doubling the estimated path length. Assuming that sediment is not trapped in the first depositional area but in the second one and doubling the path length we more closely estimate the sediment transport at the higher discharges (i.e., estimates are not significantly different than the measured averages ($p>0.05$) but overestimate the sediment transport at the 0.7 l/s and 1 l/s discharges ($p<0.05$) (Fig. A5).

**5.1.2 Confinement**

It is possible that due to the confined condition of the flumes, channel width may exert an outsized effect on the average bedload transport distance as the channel is unable to widen in response to an increase in discharge, therefore causing a flushing effect. In the flume experiment, we found that the VMD-HD method performed better at the lower discharges of 0.7 l/s and 1 l/s but significantly underestimated the sediment transport at the 1.5 l/s and 2 l/s discharges (Fig. 6). The underestimation at higher discharges could be related to the amount of morphological change relative to the sediment transport. Recently, Booker and Eaton (2022) quantitatively explored the link between sediment transport and morphology and proposed an index to represent the intuitive notion that as sediment transport increases relative to morphological change, the processes become decoupled and inferences from one to another become more difficult. They developed a 'throughput index' which is the ratio between sediment flux and morphological change and represents how much sediment moves through a reach without leaving a topographic signature of equal magnitude. Therefore, the ratio represents how well the flux is represented morphologically with the ratio approaching 1 when all of the flux is shown as morphological change and exceeding 1 when there is transport without equivalent morphological change. In our case the flume experiments were confined, therefore, as discharge increased the channel was not able to widen and deform laterally potentially causing the sediment to move through the flume without leaving an equivalent topographic signature. To explore the applicability of the method proposed we calculated the morphological active width by counting the percentage of pixels in the DoD that showed topographic change after filtering (we applied this metric only for the flume experiments since the San Juan River DoDs do not include the submerged part of the channel). The morphological active width increased with discharge as expected and was positively correlated with the error of our estimates (Fig. 8). This result exposes a limitation of the morphological method in general and our application specifically, that is, confined channels with high transport relative to morphological change are likely poor candidates for the morphological method as inferences between changes in morphology and sediment transport become decoupled. Further applications of this method in the field and in the lab could identify a potential threshold defined by the throughput index (Booker and Eaton, 2022) or the morphological active width described in this study. The advantage of using the morphological active width as opposed to the throughput index is that it can be determined from the DoD without direct sediment transport measurements.

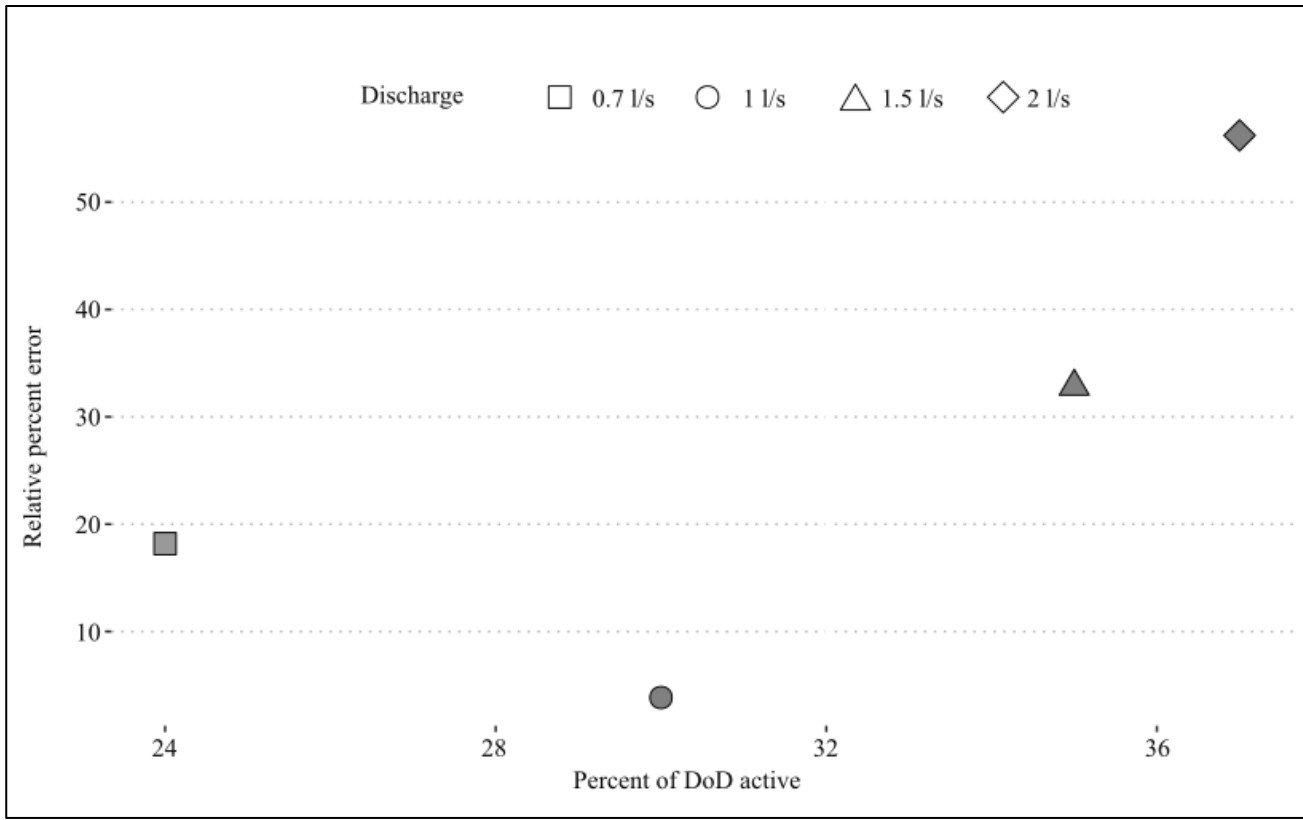

**Figure 8: Relative percent error between estimated flux using the VMD-HD method and the measured flux in the flume experiments (y-axis) vs the percentage of the DoD showing morphological change (x-axis). Different discharges are denoted by shape.**

**5.1.3 Morphological controls**

Previous studies have shown that in gravel bed rivers, macroform spacing is typically 5-7 channel widths (Montgomery and Buffington, 1997) and therefore half of that spacing, i.e. pool to bar, may be considered a proxy for the characteristic path length. We compared our estimates of path length to (half of) both 5 and 7 times the channel width in the flumes and found that the IMF 5 estimates of path length were between the 5 and 7 channel widths for all but the highest discharge (Fig. A6). Interestingly, the manually measured distances were less than the 5-7 channel widths for all discharges but approaching 5 channel widths at the 2 l/s discharge (Fig. A6). When used to calculate sediment flux, the estimates derived from using 5 and 7 channel widths were not significantly different than our VMD-HD estimates at discharges 0.7-1.5 l/s or the measured flux at all discharges (Fig. A5). Here we are likely seeing a good correspondence between the characteristic path length, width, and sediment transport, because at formative discharges, morphology is the primary control on bedload travel distance. Whereas at lower discharges, where the morphology is relatively stable, discharge may exert a stronger control on path length. Because we do not have tracer data in the flumes for comparison, we can only rely on the sediment transport measurements for validation but further flume studies with both sediment flux and tracer data for validation could help resolve this question. The periodicity we extract from the DoDs as an estimate of path length corresponds to previous observations of preferential particle deposition at specific morphological units and relationships to channel morphology (Beechie, 2001; Pyrce and Ashmore, 2003a, b; Kasprak et al., 2015; McDowell and Hassan, 2020; McDowell et al., 2021). In the San Juan River study, our estimates aligned closely with the secondary modes in the particle path length distributions (Fig. 7) consistent with observations that at channel forming flows, particle path lengths tend to be bi or multimodal with secondary modes corresponding to the

location of bars (Pyrce and Ashmore, 2003b). This preliminary result should be further examined with additional field
data in multi-threaded channel types.
We expected that the path length in more complex channels such as braided configurations would be more difficult to
estimate due to the possibility of multiple path lengths active at different flow stages. In this study both the flume
experiment and the field study exhibited a wandering morphology although in the flume experiment, the channel began
to simplify at higher discharges, likely due to the inability of the channel to widen in response to the increase in
discharge. Further, path length estimates did not change significantly between the discharges whereas the erosion
volume increases with discharge, and that, as mentioned previously, potentially contributed to the underestimation of
sediment flux at the higher discharges. Additionally, at the 1.5 l/s, and 2 l/s discharges, the patches of erosion and
deposition began to overlap, therefore, the wavelike pattern from areas of erosion to deposition represented by the IMF
5 vector became flattened (Fig. 3, A1). To disentangle the confounding erosion and deposition from the net vector, we
applied the VMD method to a vector created from erosion and deposition separately. When calculating the path length
using the erosion or deposition vectors, we took half of the resulting path length as we are still interested in the distance
from erosion to deposition rather than erosion to erosion. We found that the path lengths generated from these vectors
were not significantly different than the path lengths generated using the net vector ($p > 0.05$) (Fig. A6) nor were the
estimates of sediment transport (Fig. A5) This evidence supports the use of the net vector in this case because it appears
that erosion and deposition were similarly distributed. However, in rivers with differing morphology, perhaps braided
systems, we might suspect that erosion will be more localized than deposition which can be dispersed (Goff and
Ashmore, 1994). In these cases, using VMD to decompose the net, erosion, and deposition separately could give further
insight into how deposition and erosion are contributing to the net change. For example, deposition may contribute little
to net vector if the relative magnitude of the oscillations is small compared to erosion which tends to be more
concentrated. In addition to estimating a characteristic path length, this decomposition could give further insight into the
nature of depositional and erosional processes in a reach. We also recognize that perhaps when multiple channels are
present and active, it may be beneficial to segregate the DoD, treating each channel as a separate system and generate
multiple path length estimations and avoid compensating erosion and deposition within the cross section. Further
investigations are needed in the lab and in the field to propose robust methodologies to assess realistic ranges of path
lengths from DoD for varying river patterns.
**5.1.4 Using the IMFs**
The path length-based method for calculating sediment transport necessitates that a single path length be selected and
this is surely an oversimplification of reality. Encouragingly, the flume experiment shows that by using the VMD-HD
method to select the path length, we are able to reasonably approximate sediment transport at the lower discharges (Fig.
6). However, when applying this method to a real case study, like that of the San Juan River, it is important to consider
if the results make sense given what is known about the channel and the time and magnitude of flood events between
surveys, potentially taking into account both IMF 4 and IMF 5 to generate a range of plausible transport or path lengths.
The VMD-HD method presented here selects one of the five IMFs to be used as an estimate of path length based on the
geometric similarity, as measured by the Hausdorff distance, of the IMF to the original data vector. However, we
presume that not only does the method occasionally select an erroneous IMF (IMF 1 for example where the path length
is on the order of mm) but it also reasons that in some cases more than one IMF could represent the pattern of erosion
and deposition in the DoD or perhaps a range due to the heterogeneous nature of sediment transport. In the flume
experiment, the VMD-HD method selected the longest wavelength, IMF 5, 74% of the time and IMF 4, 24% of the
time. There were only two instances in which IMF 1 was selected and neither IMF 2 or 3 were ever selected. Likewise,
IMF 5 was selected for all three bars in the San Juan River dataset. This result agrees with observations from the signal
processing literature wherein the lower frequency (in our case wavelength) IMFs (4 and 5) are thought to represent the
true signal whereas the higher frequency (shorter wavelength) IMFs are attributed to noise (Boudraa et al., 2005). In our
case we can verify visually that IMF 5 is most likely representative of the characteristic path length by tracing the path
from erosional site to depositional site within the DoD using the manual method (Fig. 9). Here we see that the longest
IMF captures the spacing between erosional and depositional patches as estimated by other methods (Redolfi, 2014;
Vericat et al., 2017; Calle et al., 2020). This study, as the others, supports the idea that the periodic nature of erosion
and deposition can be used to estimate sediment transport and helps to clarify the conditions where this approach is
valid. Moreover, this study provides an objective and repeatable method to estimate the characteristic path length.

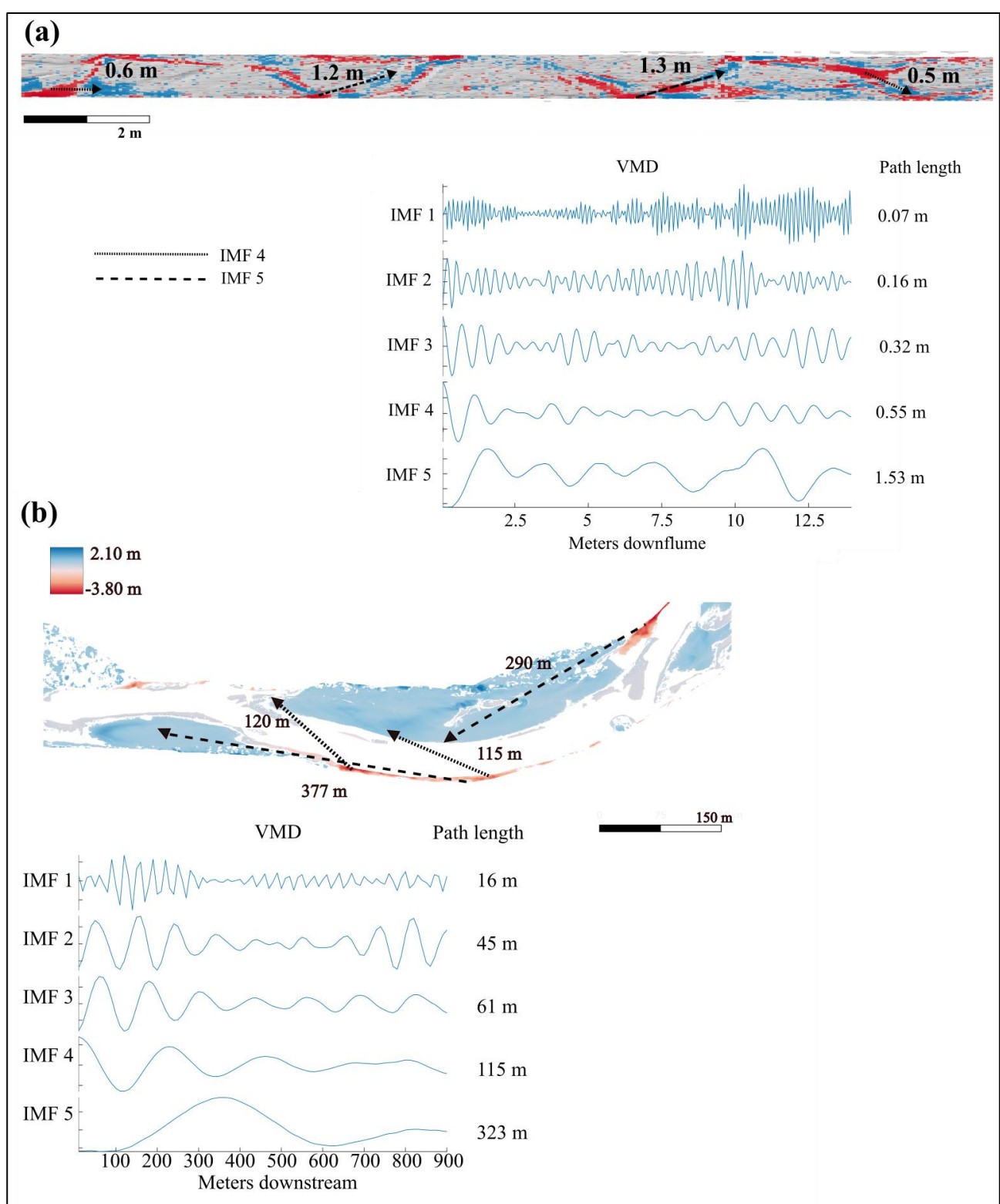

**Figure 9: DoD with arrows showing possible path lengths between areas of erosion (red) to deposition (blue) corresponding to both IMF 4 and IMF 5. The VMD breakdown including all IMFs and the corresponding path lengths are shown for an experimental run from the 1.5 l discharge (a) and bar 15 from the San Juan River (b).**

Different IMFs also allow us to explore multiple periodicities, such as shorter path lengths in the DoDs that may correspond to IMF 4 (Fig. 9). The method we present here to select one of the IMFs to represent the periodicity is convenient for assigning a characteristic path length to be used in sediment transport calculations. However, it is unclear if the range of IMFs may be used to estimate aspects of the path length distribution. As a first step, we see that in the

San Juan River the path lengths associated with IMF 4 and IMF 5 bracket the mean, median, and key depositional areas
associated with the path length distribution (Fig. 7). With future studies it may be possible to set a range of plausible
transport based on IMFs 4 and 5.
**5.2 DoD related uncertainties**
Any application of the morphological method using DoDs is sensitive to the error thresholding method used due to the
way in which different thresholding techniques influence both the volumes of erosion and deposition as well as their
spatial patterning (Brasington et al., 2003; Wheaton, 2008; Wheaton et al., 2010; Vericat et al., 2017). Because our
method relies on the spacing between areas of erosion and deposition which is related to the size of the patches as well
as which patches are detected, we considered that thresholding techniques could greatly affect the estimates of path
length. We tested this hypothesis by applying the method to both the raw and filtered DoDs for the Trento flume
experiment and found that while the volumes of erosion and deposition were lower after thresholding as expected
($p<0.001$), the path length estimates were not significantly different ($p>0.05$) (Table A1). While the thresholding here
did not affect the path length estimates, we might imagine a scenario in which an entire area of erosion or deposition is
removed through aggressive thresholding techniques, thereby potentially affecting the path length estimates and
therefore caution that appropriate thresholding is important for the application of this method and the morphological
method in general. It is also important to consider the spatial resolution (i.e., raster cell size) of the DoD when applying
this method. Similarly to thresholding or selecting a bin size, the spatial resolution of the DoD could cause information
to be lost if the cell size is large enough to aggregate erosion and deposition within the same cell (see for instance the
comparison made in Antoniazza et al., 2019). We see less of a risk in using smaller cell sizes as the method already
calls for aggregation in the binning process and in theory VMD should be able to separate the small scale fluctuations as
short wavelength IMFs. However, this is an open question and should be evaluated by the user on a case by case basis.
The time between surveys is of equal importance to the path length in the estimation of virtual velocity and in the field
can be highly uncertain due to poor availability of hydrologic data and/or the uncertainty of estimating the onset of
transport based on a critical shear stress. Further, as time between surveys increases, so too does the probability of
compensating erosion and deposition which can affect both the volumes of erosion and deposition and the topographic
signatures (Lindsay and Ashmore, 2002; Vericat et al., 2017) necessary for VMD-HD method. We tested how the time
between surveys might affect both the volumes of erosion and deposition and our path length estimates by differencing
DEMs not every time step but between two, three, and four timesteps, each time step being one of the nine runs in the
lab of phase 3 (see method). Not surprisingly the volume of erosion and deposition increased significantly with
increasing time between surveys with the largest increase between the 1[st] timestep and 2[nd] timestep (Fig. 10). The path
length estimates did not increase significantly for any of the discharges (Fig. 10c) indicating that the path length
estimate is stable, likely because, as already noted, the spacing of erosion and deposition is related to the position of
erosional and depositional features which do not change much in the flume. When both of these parameters are used in
the sediment transport calculations and normalized by the increased time between surveys, we found no statistically
significant difference between the estimates (Fig. 10d). However, though not statistically significant, there is an
apparent decreasing trend in the sediment flux with the increased time between surveys, especially for the 2 l/s
discharge that may indicate compensation (Fig. 10d). Despite the apparent trend at the highest discharge this is a
promising result in that even by increasing the time interval by a factor of 4 we are still able to estimate sediment
transport reasonably at the lower discharges. In the field there are often multiple flood events of differing magnitude in
the year between surveys as was the case with the San Juan River study (McQueen et al., 2021). Although there were
five flood events of differing magnitudes between the San Juan River surveys, we were still able to estimate path
lengths corresponding to potentially significant features of the path length distributions (Fig. 7).

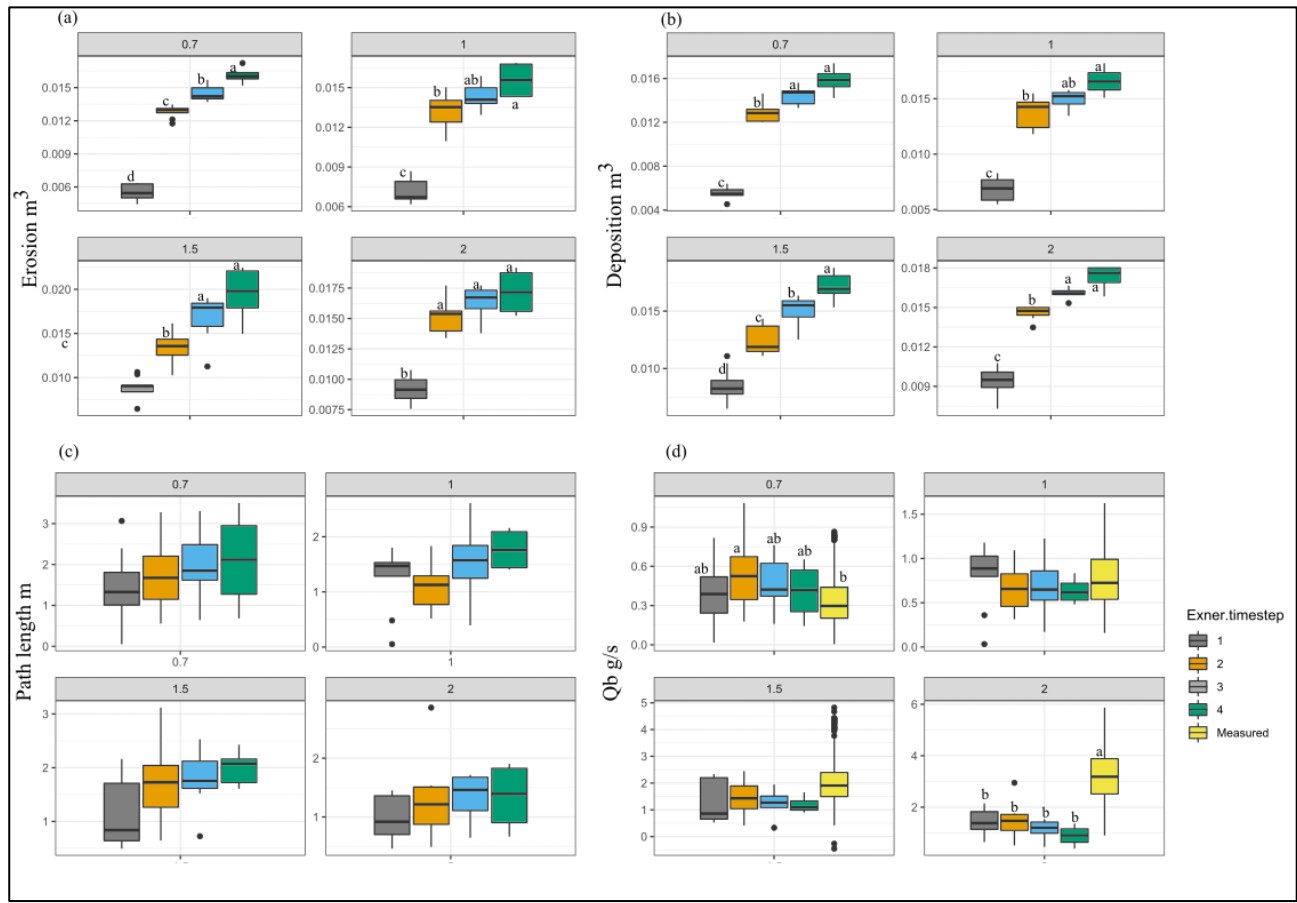


**Figure 10: (a) Erosion measured from the flume experiments for each discharge and each timestep (b) deposition**
**(c) path length estimates using VMD-HD method (d) sediment flux estimated using VMD-HD method and**
**measured. Significant post-hoc Tukey results are denoted by letters a-d ($\alpha$=0.05).**
**6 Conclusion**
Given the observed connections between morphology and path length at channel forming flows, we proposed that the
periodic nature of the pattern of erosion and deposition can be a proxy for a characteristic path length in gravel bed
rivers. We applied tools from signal processing to quantify this periodicity and found that by the longest wavelengths
from the decomposition, IMF 4 and IMF 5 may represent meaningful bedload transport processes and IMF 5 in
particular may represent the characteristic path length. We found that the path length estimates generated by IMFs 4 and
5 bracket a significant portion of measured path length distributions in the field and correspond to important
morphological units. In the flume experiment we found that IMF 4 and 5 path lengths also bracket the manually
measured distances between erosional and depositional patches and when extended to calculate sediment flux our
estimates were not significantly different from the measured average at low discharges. Importantly we found an
insensitivity of the method to increasing discharge and propose that perhaps limits arise where discharge increases in
confined settings, such as in the flume, and sediment transport becomes decoupled from morphological changes. Our
method provides a new view of the periodic nature of erosion and deposition in sediment transport and a novel way to
extract sediment transport information using only DoDs.


**Appendix A**

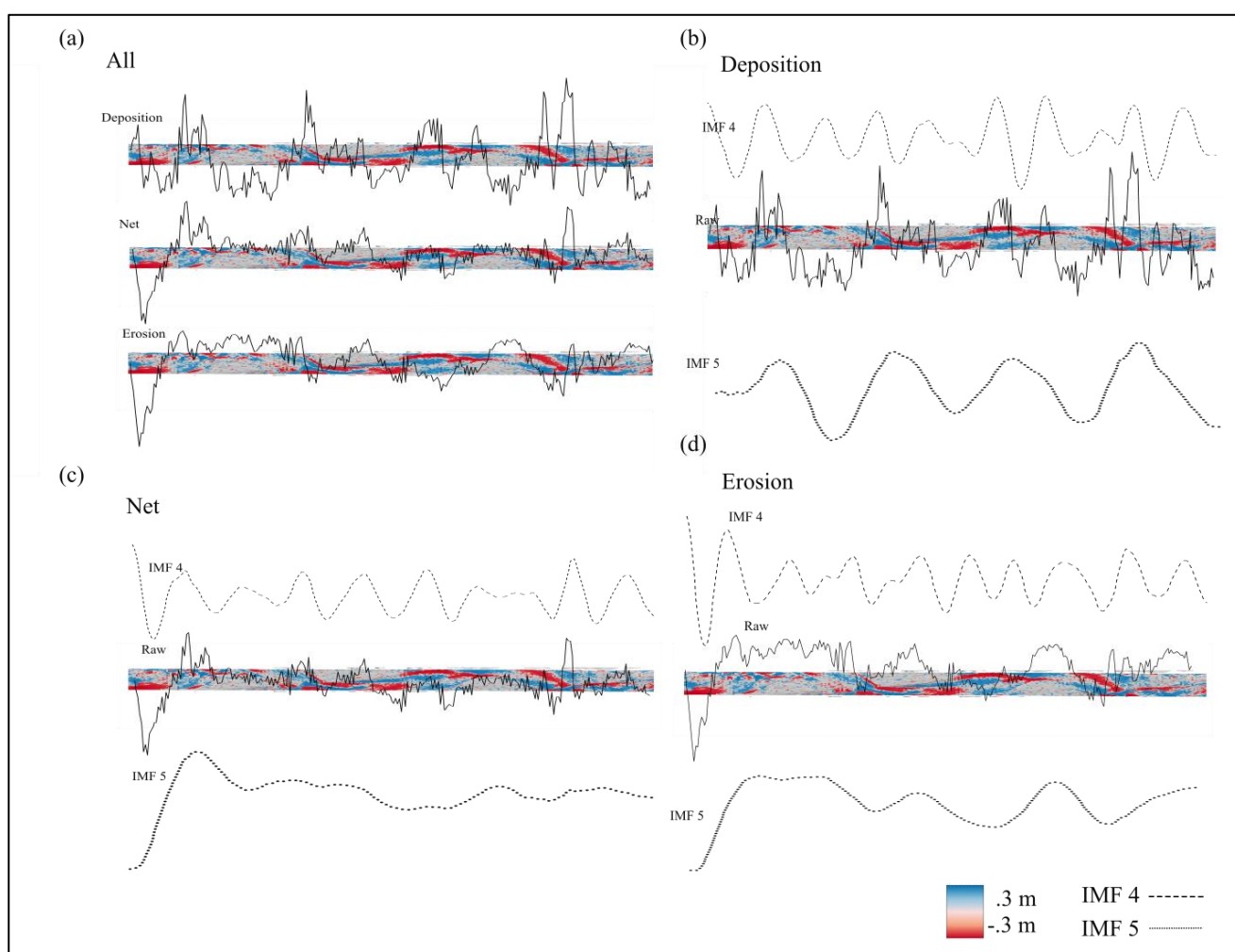

Figure A1. DoDs from the 2 l/s discharge. a) Vector of deposition, erosion, and the net. b) Raw depositional vector and the decomposition of IMF 4 and IMF 5 from that depositional vector. c) Net vector and the decomposition of IMF 4 and IMF 5 from that net vector. d) Raw erosional vector and the decomposition of IMF 4 and IMF 5 from that erosional vector.

660

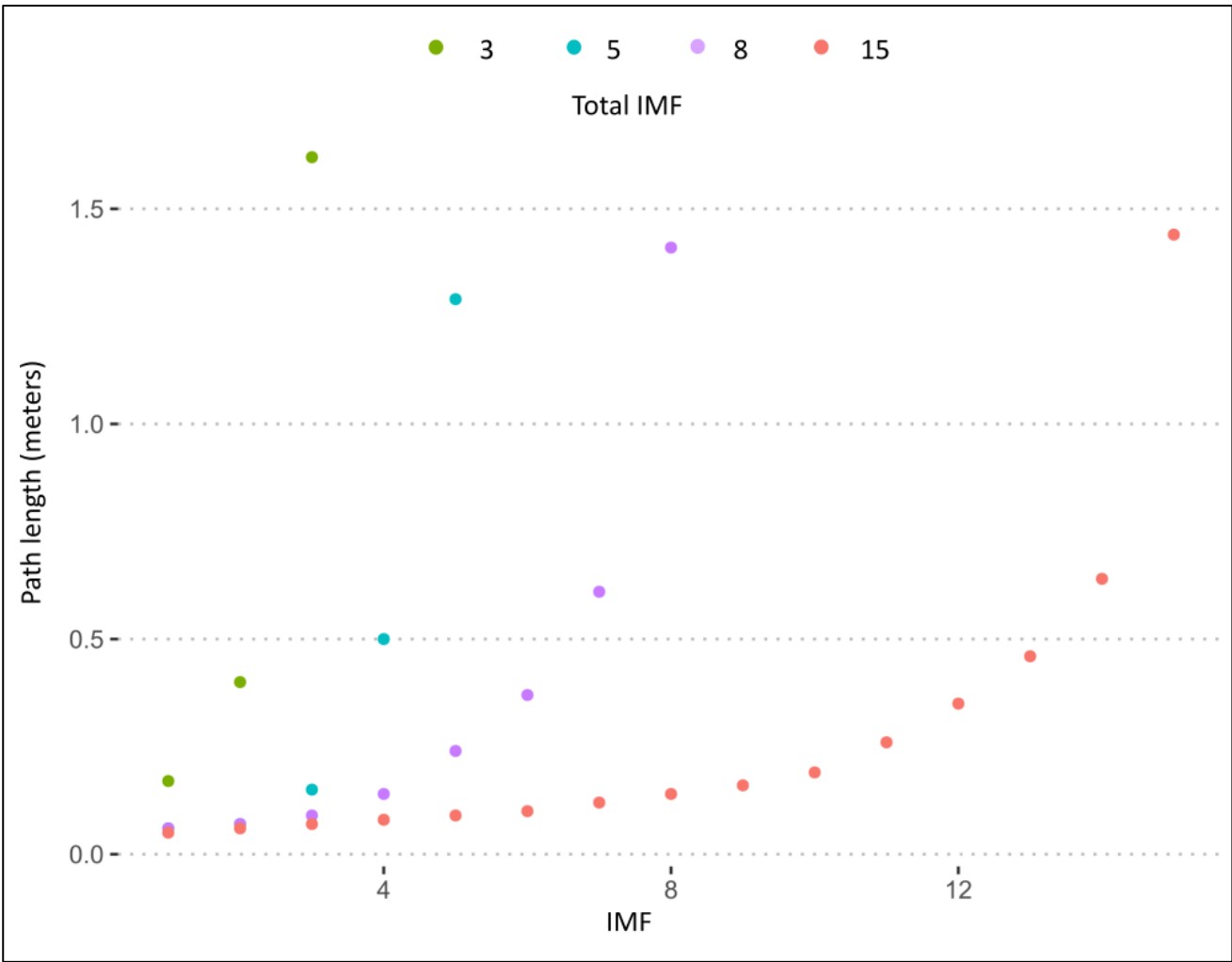

Figure A2. Path length estimates using a maximum of 3,5,8, or 15 IMFs.

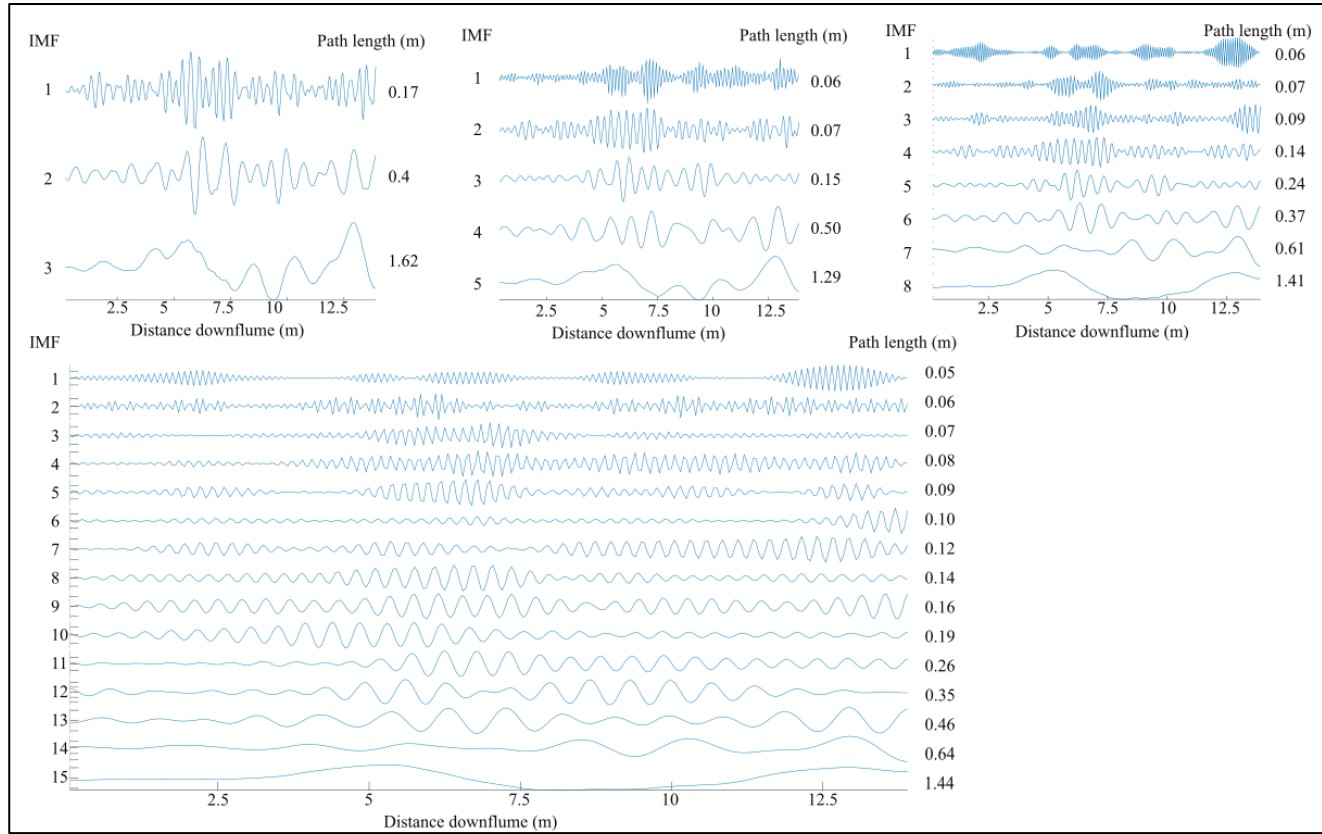

Figure A3. Path length estimates from VMD for 1.5 l/s discharge. Sensitivity of maximum number of IMFs.

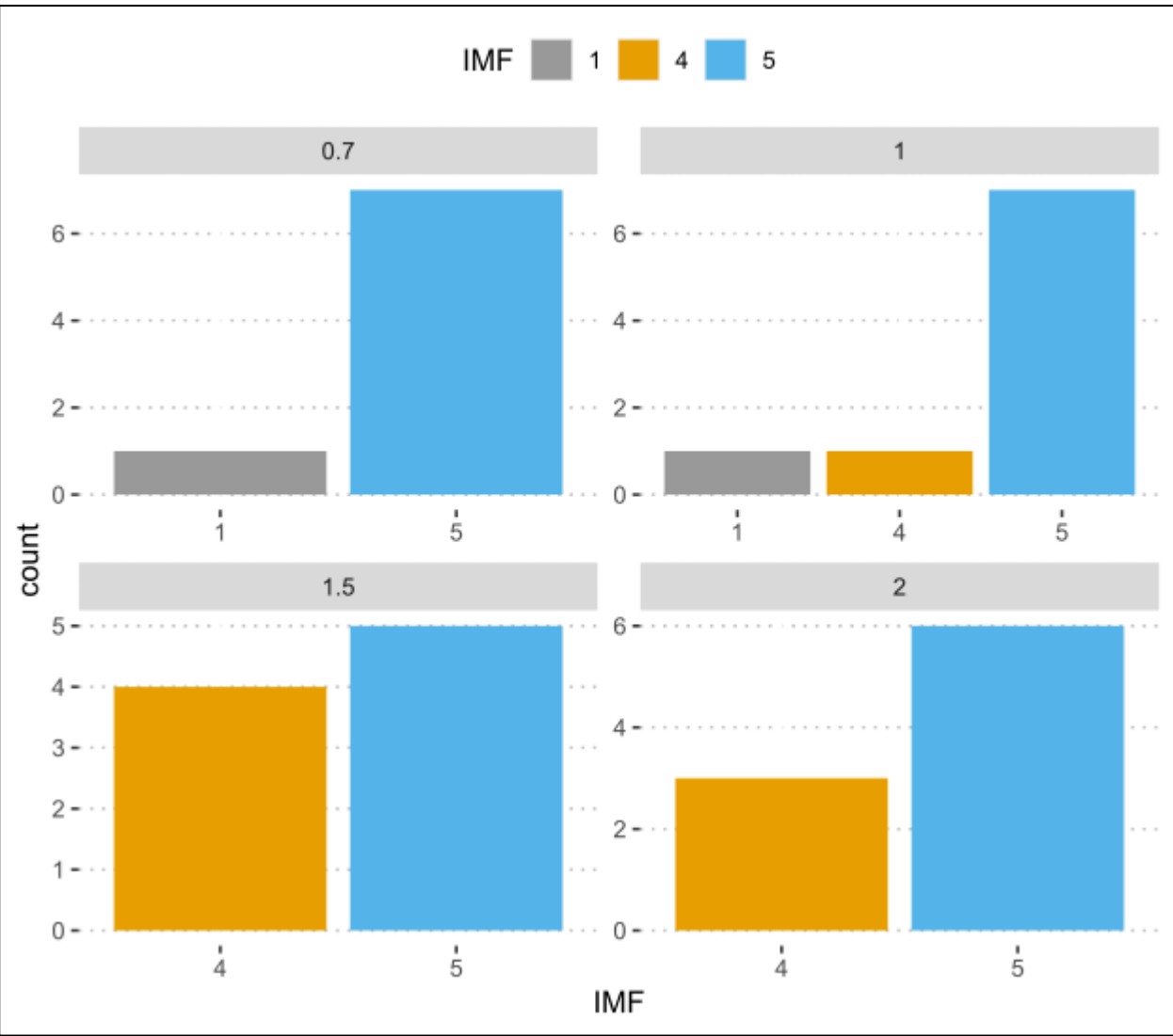

Figure A4. Number of times each IMF was selected by the VMD-HD method for each discharge.

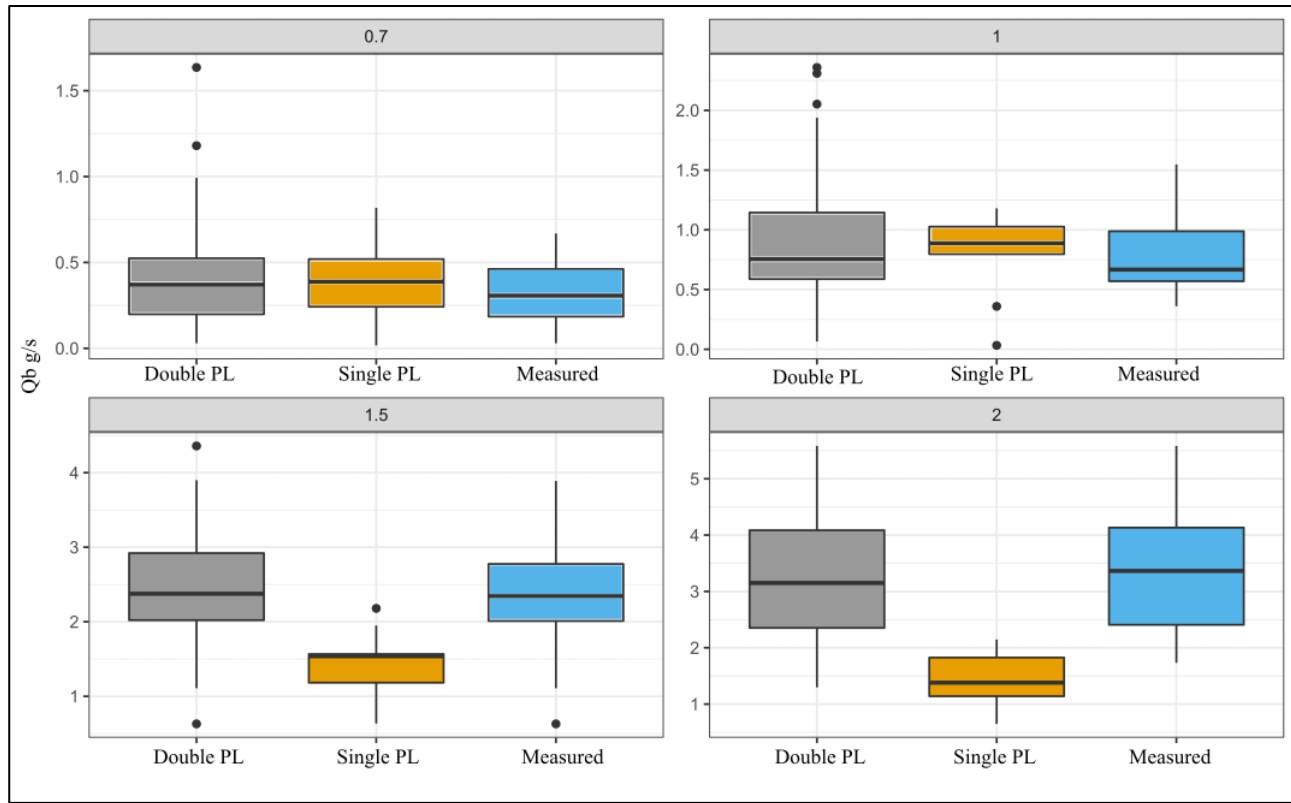


Figure A5. Sediment transport calculated using the single path length estimate from the VMD-HD method (b) and
doubling the path length estimate (a). Estimated flux is red and measured flux is blue. Significant p values are shown.

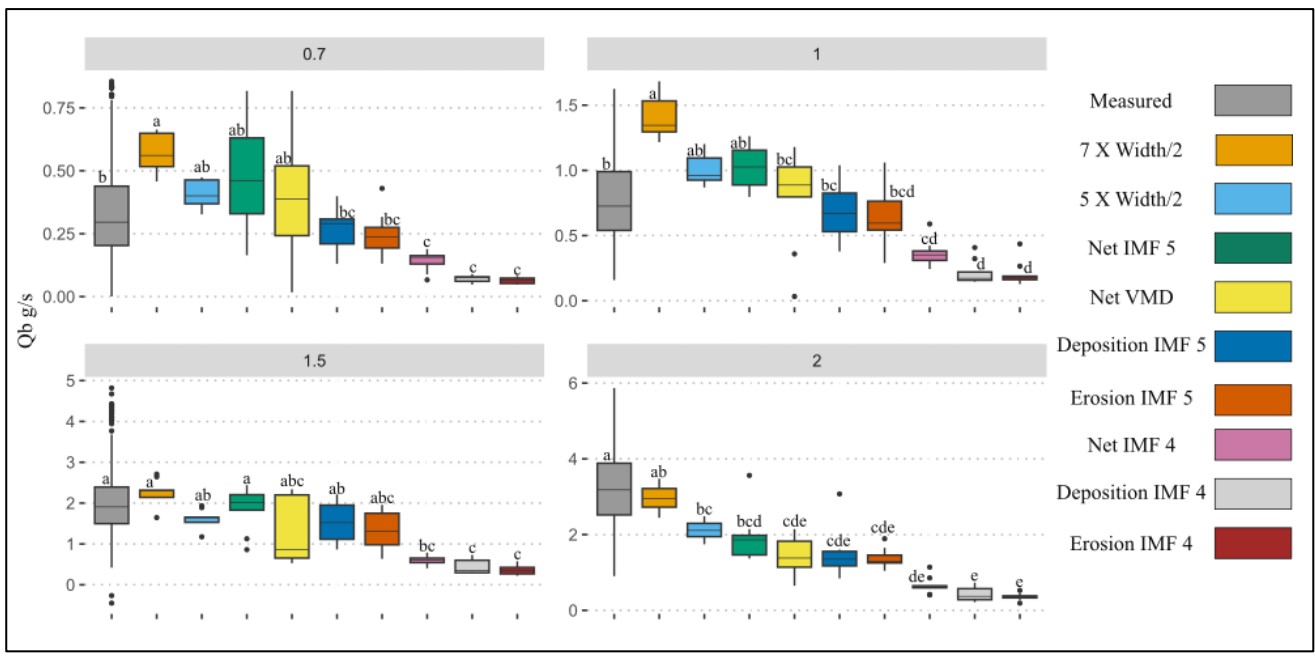


Figure A6. Sediment transport (g/s) calculated using channel dimensions, IMFs 4 and 5 for net, erosion, and deposition
vectors. Compared to the measured flux for each discharge. Post hoc Tukey results denoted by letters a-f.

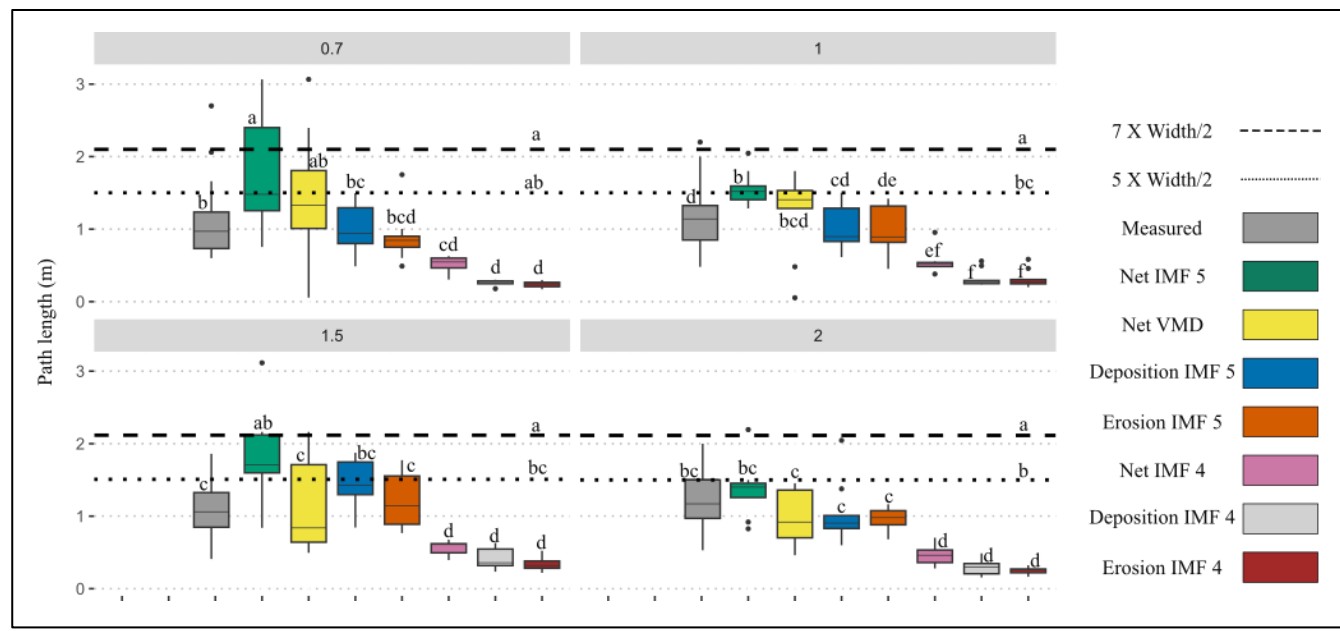


Figure A7. Path length estimates from the channel dimensions, IMFs 4 and 5 for net, erosion, and deposition vectors
compared to manually measured distances for each discharge. Post hoc Tukey results denoted by letters a-f.
Table A1. Results from filtered vs raw DoDs from the flume experiments.

| Discharge | Path length raw (m) | Path length filtered (m) | Qb estimated raw(g/s) | Qb estimated filtered (g/s) | Erosion raw (m3) | Deposition raw(m3) | Erosion filtered (m3) | Deposition filtered (m3) |
|---|---|---|---|---|---|---|---|---|
| 0.7 | 1.77 | 1.31 | 0.69 | 0.30 | 0.01 | 0.01 | 0.01 | 0.01 |
| 0.7 | 0.80 | 0.75 | 0.29 | 0.14 | 0.01 | 0.01 | 0.00 | 0.00 |
| 0.7 | 3.05 | 3.06 | 1.20 | 0.71 | 0.01 | 0.01 | 0.01 | 0.01 |
| 0.7 | 2.54 | 2.40 | 0.97 | 0.51 | 0.01 | 0.01 | 0.01 | 0.01 |
| 0.7 | 2.30 | 0.05 | 1.01 | 0.01 | 0.01 | 0.01 | 0.01 | 0.01 |
| 0.7 | 0.87 | 1.09 | 0.35 | 0.23 | 0.01 | 0.01 | 0.01 | 0.01 |
| 0.7 | 1.57 | 1.61 | 0.70 | 0.43 | 0.01 | 0.01 | 0.01 | 0.01 |
| 0.7 | 1.24 | 1.35 | 0.56 | 0.37 | 0.01 | 0.01 | 0.01 | 0.01 |
| 1 | 1.25 | 1.41 | 1.04 | 0.89 | 0.01 | 0.01 | 0.01 | 0.01 |
| 1 | 1.37 | 0.48 | 1.28 | 0.31 | 0.01 | 0.01 | 0.01 | 0.01 |
| 1 | 1.25 | 1.47 | 1.10 | 1.02 | 0.01 | 0.01 | 0.01 | 0.01 |
| 1 | 1.23 | 1.80 | 1.18 | 1.00 | 0.01 | 0.01 | 0.01 | 0.01 |
| 1 | 1.48 | 1.59 | 1.46 | 0.84 | 0.01 | 0.01 | 0.01 | 0.01 |
| 1 | 1.83 | 1.53 | 1.57 | 0.77 | 0.01 | 0.01 | 0.01 | 0.01 |
| 1 | 1.54 | 1.52 | 1.27 | 0.75 | 0.01 | 0.01 | 0.01 | 0.01 |
| 1 | 1.51 | 1.29 | 1.21 | 0.69 | 0.01 | 0.01 | 0.01 | 0.01 |
| 1.5 | 1.12 | 0.68 | 1.82 | 0.53 | 0.01 | 0.01 | 0.01 | 0.01 |
| 1.5 | 1.63 | 0.84 | 2.30 | 0.86 | 0.01 | 0.01 | 0.01 | 0.01 |
| 1.5 | 1.50 | 0.64 | 2.74 | 0.65 | 0.02 | 0.01 | 0.01 | 0.01 |
| 1.5 | 0.50 | 0.61 | 0.82 | 0.67 | 0.01 | 0.01 | 0.01 | 0.01 |
| 1.5 | 0.85 | 0.49 | 1.41 | 0.54 | 0.01 | 0.01 | 0.01 | 0.01 |
| 1.5 | 0.51 | 1.60 | 0.76 | 2.01 | 0.01 | 0.01 | 0.01 | 0.01 |
| 1.5 | 1.67 | 1.71 | 2.28 | 2.21 | 0.01 | 0.01 | 0.01 | 0.01 |
| 1.5 | 1.50 | 2.16 | 2.54 | 2.20 | 0.01 | 0.01 | 0.01 | 0.01 |

| 1.5 | 1.13 | 2.12 | 1.73 | 2.34 | 0.01 | 0.01 | 0.01 | 0.01 |
| 2 | 1.41 | 1.40 | 2.53 | 1.86 | 0.01 | 0.01 | 0.01 | 0.01 |
| 2 | 0.91 | 0.92 | 1.58 | 1.19 | 0.01 | 0.01 | 0.01 | 0.01 |
| 2 | 1.26 | 1.36 | 2.05 | 1.58 | 0.01 | 0.01 | 0.01 | 0.01 |
| 2 | 1.13 | 1.26 | 1.67 | 1.27 | 0.01 | 0.01 | 0.01 | 0.01 |
| 2 | 0.06 | 1.45 | 0.09 | 1.61 | 0.01 | 0.01 | 0.01 | 0.01 |
| 2 | 0.46 | 0.46 | 0.78 | 0.56 | 0.01 | 0.01 | 0.01 | 0.01 |
| 2 | 1.32 | 0.83 | 2.04 | 1.18 | 0.01 | 0.01 | 0.01 | 0.01 |
| 2 | 0.71 | 0.66 | 1.30 | 0.74 | 0.01 | 0.01 | 0.01 | 0.01 |
| Summary | | | | | | | | |
| Discharge | Erosion | Deposition | Path Length | Qb | | | | |
| 0.7 | $p<0.001$** | $p<0.001$** | $p>0.05$ | $p<0.05$* | | | | |
| 1 | $p<0.001$** | $p<0.001$** | $p>0.05$ | $p<0.05$* | | | | |
| 1.5 | $p<0.001$** | $p<0.001$** | $p>0.05$ | $p<0.05$* | | | | |
| 2 | $p<0.001$** | $p<0.001$** | $p>0.05$ | $p>0.05$ | | | | |
| | | | | | | | | |
| *p-values from student's t test between raw and filtered data | | | | | | | | |


## Data availability
Data is available at https://doi.org/10.5281/zenodo.8014453.

## Author contribution
LC, SB, WB and NS conceptualized the study. EP preformed the experiments. LC, SB and WB designed the method. LC performed statistical analysis. LC, EP, and WB wrote the manuscript. LC, SB, EP, WB, and NS edited the manuscript.

## Competing interests
The authors declare they have no competing interests.

## Financial support
This work was supported by the CARIPARO foundation and the University of Padova.

## Acknowledgements
We would like to thank the three anonymous referees for their time and thoughtful suggestions which have considerably improved the quality of this work. We would also like to thank the CARIPARO foundation and the University of Padova.

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
