# Peer review of "Path length and sediment transport estimation from DEMs of"

_EGUsphere, 2023_

## Author Comment (AC1)

**This comment is in response to anonymous reviewer 1 (RC1).**
**Thank you very much for taking the time to thoughtfully review our manuscript. We have worked to carefully consider all of your proposed suggestions and revisions. Below we will give responses to the specific questions.**

This paper provided a signal processing approach for estimating path length and sediment transport from DoDs. This topoic is intresting and suible for ESurf. However, there are several substantial issues.
First, the reasearch gaps are not clear in the Introdution section. More literatures should be reviewed. For example, signal processing approaches is the main method of this paper, but it is not reviewed in the introduction section. Why should we need the signal processing approaches? Is other method also OK? namely, what is the necessarity of this approach. In fact, the signal processing approaches, such EMD method, are already applied in geoscience (e.g. DOI: 10.1111/tgis.12512).

**Author's response- We have expanded the introduction to highlight the gaps we are trying to fill as well as explain the justification and rationale for using the signal processing method. (See extensive changes to the introduction and Sect. 2.1 Path length).**

Second, the review of the morphological method is incomplete. The morphological method including 1D and 2D approaches is widely used for infering sediment transport (DOI: 10.1002/esp.483; DOI: 10.1002/esp.4633; DOI: 10.1002/esp.5094; doi.org/10.1016/j.catena.2022.106244). With these method, infering sediment pathway is easy. Although they did not estimate the path length, their methods have the potentaility for path length.

**Author's response- We agree that these are important studies to include and have expanded the introduction to include some of these articles and much more literature on applications of the morphological method. (See introduction section, extensive changes, ex. Lines 27-37 and 103-119).**

I suggest to re-orginize the Introduction, and review more relative literature.
Third, the methodology is not rigorous.
1. The manual measurement of the spacing is with huge uncertainty.
2.the center to center measurement seems OK, but, notably, the path length may be false. Besides the nearst patch of deposition to erosion patches, the volum of deposition and erosion pathes must be considered. If the volum of deposition greater than the volum of erosion, the sediment will be further transported to downstream patches; and thus the path length is not the nearest spacing of erosion and deposition patches.

**Author's response- This is a good point and we have noted that this might be what is happening at the two higher discharges. We have included these limitations in the explanation of the manual method (lines 144-146). We have expanded the discussion to include a discussion of the vectors of erosion and deposition separately as well as the net and how they may be out of balance (lines 361-375 Fig. A1)**

3. How do you determine the direction and size of the "bins" (L107). It is important because this affects the sum of the DoD matrix, and then influence the results. Moreover, the effect of direction and size of the bins should be disscussed.

**Author's response- We agree and have expanded the methods section to discuss bin size selection (see lines 171-178 and lines 503 in the discussion).**

Fourth, the figures should be improved. For example, Fig.2 a lack of legend. Fig.3 the IMF5 lack of axis.

**Author's response- Thank you for pointing this out. We have now fixed the figures.**

Finally, this paper segemented DoD into bins. Then, it is a 1D method in fact. with the 2D methods for sediment routing were already developed. Why not try the 2D methods. the 2D sediment tranport path length should be more correct than the 1D path length.

**Author's response- Thank you for this suggestion. It would have been interesting to explore this aspect. However, in the case of our study, it seems slightly out of the scope because our goal was to create a method to define a characteristic path length to be used in sediment transport calculations solely from the DoD. The 2D approaches are useful but they also require either hydrologic data or estimates that are not always available. We think our method is an alternative to 2D methods when hydrologic data is not available or desired.**

---

## Author Comment (AC2)

**This comment is in response to anonymous reviewer 2 (RC2).**
**Thank you very much for taking the time to thoughtfully review our manuscript. We have worked to carefully consider all of your proposed suggestions and revisions. Below we will give responses to the specific questions.**

Thank you for the opportunity to review this manuscript. The manuscript uses a signal processing approach to estimate a representative path length for reach-averaged sediment transport estimation from a DEM of Difference. The topic is interesting and suitable for Earth Surface Dynamics. I am impressed with the method proposed in this paper. I believe it to be creative and containing the potential to explore sediment transfers from a DoD; however, I believe that this paper is currently unsuitable for publication. I believe this for several reasons.

First, the paper does not adequately provide sufficient background about the chosen topic. For example, it does not meaningfully define a path length, nor does it engage with the complexity of reducing a distribution of path lengths to a single number (i.e., "a characteristic path length"). The discussion of morphologic methods for bedload transport estimation and other methods to estimate path lengths from a DoD should also be expanded in the introduction. This lack of background information prevents the paper from making a sufficient case for why an abstracted approach to path length estimation, like the one presented here, is required. This is particularly true, as implicitly the paper uses the "manual method", which is far more concrete and can be automated, to validate their approach. There is likely a good reason for the signal processing approach, but the case needs to be made.

Second, the Methods provided are not sufficient to allow other researchers to adopt the procedure. It needs more detailed explanations of the decisions made, guidance for researchers who may wish to apply this method across river environments, and an in-depth discussion of the assumptions and simplifications inherent in the method (for example, I find the discussion in Lines 307-323 both too surficial and coming far too late in the paper).

Third, it does not seem obvious to me that the paper tests the stated hypothesis that path length can be inferred from changes in morphology. Some of this is a definitional issue, where the lack of clarity in the term "path length" (versus the undefined "characteristic path length") muddles the paper. In the field scenario, the paper compares the morphologic estimate of path lengths with tracer path lengths, whereas in the lab scenario, the paper compares the reach-averaged bedload transport estimated with the morphologic method against a measured bedload flux. This is a different comparison, as the morphologic method presented in Equation 3 also contains terms other than the path length (mainly the measured volume of erosion). This might not be a problem if either the true volumes of erosion (i.e., including compensation) or the displacement distances of particles (using tracers – as difficult as they are to apply in a stream table) could be more clearly constrained. I imagine the erosion volumes (including throughput) could be constrained with a full sediment budget for each experimental period, which might help with linking the morphologic signature to the transport processes (and be used to strengthen the discussion of the throughput index).

Finally, I know that this recommendation might be beyond the scope of the paper presented, but I encourage the authors to consider ways to incorporate the more complete output of the signal processing approach (i.e., more of the IMFs) into descriptions of sediment transport processes.

The signal processing approach seems like it may be a powerful way to describe the heterogeneity in bedload transport processes and encourage its further development and exploration.

Below, I highlight places that the manuscript could be improved:

Introduction:

1. This paper requires a clearer definition of path lengths. For example:

    1. The initial definition of path length (line 25) is "how far sediment travels". This wording seems to read that there is a single path length that describes the entire study segment. Later in the paper (line 322) the authors claim "in reality there is not one path length but rather a distribution." These two statements seem to conflict. Generally, I've always understood "path length" to describe the travel distance of one grain during an event (for which a population can be described through a distribution). Here, the authors seem to use the term "path length" to describe a characteristic path length. This is not a problem, but it should be explicitly stated.

        **Author's response- This is an important point that we have now tried to clarify. We are referring to the characteristic path length and have expanded the introduction significantly to discuss the differences in path length terminology and clarify what we are aiming to estimate. We have also included further discussion of this topic in the discussion (See lines 60-67 and 78-90 and sect. 5.1.4 Using the IMFs)**

    2. Also, the reader would benefit from some discussion of the characteristic path length. How does the characteristic path length relate to a hypothetical path length distribution as measured by tracers? Is it the mean? The mode? Or is it just a value that makes the morphological method "work"? This is particularly important as the paper estimates a characteristic path length and compares the estimate against tracer displacements and bedload transport measurements.

        **Author's response- Absolutely! We think this is an important topic for discussion also in light of our study (See lines 5.1.3 Morphological controls and sect. 5.1.4 Using the IMFs).**

2. The review of previously presented methods to estimate a characteristic path length from a DoD is unsubstantial. The authors briefly mention two methods 1) the pairing of two morphological units from Neill (1971) and 2) The pairing of zones of erosion and deposition discussed in Section 2.2 (although this subsection requires citations). Other sediment-budget based methods have been presented as well. Calle et al (2020) (DOI: 10.1002/esp.4765) and McDowell et al (2021) both present ways that sediment budgets can be used to estimate characteristic path lengths.

    1. Extension of this review is important as the manuscript does not highlight the need for their presented method. Under what conditions is the somewhat abstracted

approach presented here an advantage over the more concrete approaches that have been used previously?

**Author's response- We agree that the review should be expanded and have included both Calle et al 2020 and McDowell et al 2021 as well as other studies in the introduction. (See extensive changes to the introduction).  We appreciate the value of these previous approaches and are not so much presenting the signal processing method as being a superior approach but rather an alternative way to view the erosion and depositional patterns by decomposing the overall pattern into its composite parts (the IMFs). We believe that the IMFs provide the potential to expand the understanding of the heterogeneity of sediment transport and how it relates to a characteristic path length of bulk sediment transport and have tried to make that clearer in the manuscript.**

3.  I think the paper would benefit from improved clarity in the paragraph starting on Line 50. For example:

    1.  It does not seem obvious to me that the paper tests the hypothesis that path length can be inferred from changes in morphology. It seems to be testing whether the signal processing approach can adequately represent a characteristic path length (whereas path length being inferred from morphologic changes seems to be a core assumption of the hypothesized method).

    **Author's response- Yes, you are correct, and thank you for bringing this to our attention. We have reworded this section to clarify our objectives and the underlying assumptions/hypotheses on which they are based. (See lines 120-128).**

    2.  What is an "event" in Line 51?

    **Author's response- We are referring to a competent flow event. We have clarified this in the manuscript (see line 121-122).**

    3.  The objectives should mention a signal processing approach.

    **Author's response- We agree and have reworded the objectives (See lines 120-128).**

Methods:

Broadly, I feel that the methods section would benefit from some expansion and revision.

1.  Lines 80-82 – These requirements are not explicitly checked for in this paper. In the field scenario, I am unsure how one would do this so its exclusion is fine, however, in the laboratory one, these requirements should be confirmed.

    **Author's response- We aimed to do this by using the time scale for morphologic evolution as the time step between DEM acquisitions (see lines 257-263 and 257-276).**

2.  Section 2.2 is a description of an application of paired erosion and deposition zones. This section needs citations as the method is not original. It also needs more methodological

detail. For example: How was the middle of each patch determined? The statement "use our knowledge of morphological processes to make a best estimate" should be expanded past the example provided in Line 94 and supported with citations.

**Author's response- We have expanded this section to explain the basis for our reasoning and the previous work on which it was based (see lines 141-161).**

3. Lines 95-97 seem to be the argument for the automated signal processing method by claiming that the "pairing" method is subjective. Would a script that automates this be similarly subjective? Erosional and depositional patches could be found with an area greater than a particular value, the center of mass of each patch could be identified, and the distance from erosional to depositional centers could be measured. Why choose a signal processing approach for automation over the one I just described? To be clear, I'm sure there are good reasons to do so, but the paper needs to clearly make that case.

**Author's response- We have expanded the section on the manual method to clarify why we are using it as a comparison and its strengths and weaknesses (see lines 158-161).**

4. Figure 1:

    1. Can a flow direction arrow be added?

    **Author's response- Yes, we have added a flow direction arrow.**

    2. Panels A and B seem superfluous.

    **Author's response- Agreed. They have been removed.**

    3. Based on my reading of the paper, it seems to contend that the characteristic path length estimate applies to the entire study reach? Is that the case? Does the estimated path length in Fig 1c also apply to downstream erosion/deposition pairs?

    **Author's response- Yes, we have updated the figure to show additional examples of what might be the characteristic path length**

5. The description of the signal processing approach in Section 2.3 needs more detail:

    1. A brief overview or methodological description of signal processing approaches in general might help researchers who are interested in the method but are not familiar with the toolset.

    **Author's response- Great point. We have now expanded this section to include more information on other signal processing approaches and the VMD-HD approach in particular (see lines 198- 210).**

    2. What should users consider when choosing the bin size? The bin size greatly impacts the results of signal processing approaches.

    **Author's response- Definitely! We have added information to help users choose a bin size appropriate for their study by citing Calle et al. 2020 and the criteria they used (see**

**lines 171-178). As well as a discussion of the risks of setting too large a bin size (lines 502-503 in the discussion).**

3. Similarly, how does the raster cell size impact the analysis? The method uses summed elevation change in each bin instead of volume so the cell size might matter.

**Author's response- Very true. We did not experiment with raster size but we can imagine that a larger cell size would aggregate change and therefore have a similar effect as underbinning. These are decisions that need to be addressed in any change detection using DEMs. We have added this in the discussion (see lines 492-497).**

4. Can you expand the reasoning for the choice of VMD over other approaches (line 122)? Expansion of the general description of VMD would be helpful for lay users (pushing the mathematical description to other citations seems appropriate).

**Author's response- Yes. We have added a brief overview of the advantages of VMD and pointed readers to a quantitative review (see lines 198-242).**

5. Why are there only 5 IMFs? Would this approach work with 3 or 7?

**Author's response- Thank you for raising this question. This is discussed in the signal processing literature and we cite those studies and have conducted a crude sensitivity analysis using 3, 5, 8, 15, and 25 IMFs for the calculations. We see that 3 appears to be too few because the longer wavelengths are not present. Whereas, more than 5 increases the number of short wavelength IMFs but does not drastically affect the wavelengths we consider to be physically meaningful (on the range of cm to m as opposed to mm). We have included this in the manuscript (see lines 226-242) and the supplementary information (Figures A2 and A3).**

6. Are wavelength and period the same thing in this section? It should just be wavelength (because there is no time domain), correct?

**Author's response- Yes correct. We have clarified the language (see line 208).**

7. More specificity with language would be helpful in section 2.3. What do each of the 5 IMFs specify (what is the y-axis in Figure 2c)? What are the authors computing the PDF of in line 129 (there seems to be a probability and a downstream distance, but I am unsure what the probability refers to, although I can guess)? What is the "original data vector" (Line 130)? What smoothing was applied? Is it the measured net elevation change in each bin?

**Author's response- We apologize for the lack of clarity here. We have added the axis information to the figures. Yes, the original net vector is the net elevation change in each bin. We have clarified this language (see line 211). The smoothing was only applied when calculating the PDFs and it was a kernel density smoothing (line 214) after Ma et al., 2017.**

8. The output path length estimate would be for the whole DoD, correct? How suitable is one path length estimate for reaches with geologic/geomorphic controls? I guess

the broader statement is there needs to be some guidance for users for evaluating whether a study extent is suitable.

**Author's response- Yes, the path length is for the whole DoD. In the discussion we mention potentially segmenting the DoD when multiple channels are present (lines 549-553). We are not sure how geologic/geomorphic controls may change the results but the assumption is that if there is forced deposition from boulders, wood, etc. It would still be captured on the DoD and therefore taken into account with the method. This may be an advantage of our method over just using channel dimensions for example. However, as pointed out in the text, a proper answer to this question requires additional applications of the method in other contexts in future research.**

6. Figure 2:
    1. Panel A: Add a flow direction arrow, key, colormap. What is the source of this image?

    **Author's response- Thank you. We have now added these items. The image is an original orthophoto and that is now stated in the description.**

    2. Panel B:
        1. Left panel – the y-axis doesn't seem to be elevation change (is it elevation?). The x-axis seems to be distance from outlet, not distance downstream (this applies to all panels)

    **Author's response- The y axis is the sum of the elevation change in a bin. The x axis is the length of the DoD starting at 0 for the most upstream section. These have been clarified in the figure and description.**

        2. Right panel – What are the units? Is it summed elevation change?

        **Author's response- Yes, the scale is off after the line is detrended to go through 0.**

    3. Panel C: What is the y-axis?

    **Author's response- They are the decomposed IMFs. The values are unique for each panel and are the central frequencies that we then convert to path length. We felt this would not be useful to include in the conceptual figure.**

7. Line 133 – Is the claim that erosion and deposition that does not align with longer wavelengths just noise? Is there an assumption that erosion and deposition is regularly spaced?

    **Author's response-Yes, we assume that the majority of erosional and depositional sites are somewhat regularly spaced. We realize that this is not always the case, especially with external forcing (i.e. large wood, boulders, manmade obstructions) but that assumption does underline the idea of using the periodicity of erosion and deposition as**

**a proxy for the characteristic path length. It is the assumption that the shortest wavelengths are associated with noise (i.e. IMF 1 and 2) because they correspond to very small distances relatively both for the flumes and field but as mentioned by this reviewer (comment 8), the range of IMFs may represent different aspects of the geomorphic system. We have edited the manuscript to include a specific discussion of IMF 4 and IMF 5 and what they might represent with respect to sediment transport and the physical path length distribution. See extensive changes to the manuscript (Sect. 5.1.4 Using the IMFs)**

8. A broad comment - by only choosing one IMF, it seems that lots of potentially useful data are being discarded. I wonder if the smaller frequency IMFs can be used to improve descriptions of the geomorphic system.

   **Author's response- Yes, we suspect the same! See previous comment.**

Flume and field data

1. Lines 139 through 144 – I'd make explicit that there are four flume experiments and three field sites (i.e., "three separate bars" is three separate sites).

   **Author's response- We have updated the manuscript (see lines 256-257 and 248-251).**

2. Line 149-150 – Can you describe the planform morphologies?

   **Author's response- We have added this information in Table 1.**

3. Was there sediment feed for the flume studies?

   **Author's response- Yes, we added a sentence in the description of the laboratory experiments with the details of the sediment feed (see lines 257-259).**

4. Line 166-168 – Why is it important that the flume experiments have similar volumes of erosion and deposition in each run?

   **Author's response- We removed this sentence as it was misleading. The use of T_ex is a way to normalize time and therefore have a similar temporal evolution across different runs.**

5. Section 3.2 – Include a paragraph about the San Juan River, its morphology, flooding regime, etc.

   **Author's response- We have added this. See lines 287-293**

6. Spell out standard deviation the first time you use it (so readers know what SD means).

   **Author's response- Fixed. Thank you for pointing this out.**

7. Line 190-192 – I understand that paired topographic and tracer data are difficult to find, so the San Juan, despite its limitations is a good choice; however, I believe that this claim needs some expansion. How much of the area is submerged? How much area do the changing

water levels impact? Can you give the readers an idea of the scale of uncertainty this limitation provides?

**Author's response- We agree and have updated the text to give justification for the use of this data with the given limitations (see lines 297-310).**

8. Line 211 – Include an equation for SMAPE – also explain why SMAPE was chosen over the more commonly used MAPE.

**Author's response- Thank you for raising this point. After consideration, we have decided to use the relative percent error and feel this is more appropriate (see lines 333-337).**

9. Line 213 – How? Which metrics are being compared?

**Author's response- After deliberation, we have decided to use the San Juan data as a qualitative comparison to see how the characteristic path length compares to the physical path length distributions. We felt that the error metrics were not appropriate because we are aiming to estimate the characteristic path length, not a distribution.**

10. A general note, the San Juan River has significant amounts of sand, whereas the tracers were gravels (ranging from 3-9 cm in diameter, I believe). If sand is making up a significant portion of the volumetric change, one would expect morphologic methods to overpredict transport lengths.

**Author's response- Yes, great point! This is especially the case with bar 15 which had a higher proportion of sand. We have now added this information to the paper (see lines 414-416).**

Results

1. Figure 3 – What is the y-axis on the shown IMF? Have you tried plotting a y-axis (with labels) on the figure?

**Author's response- We have remade this figure including a y-axis.**

   1. Are there manually derived distances for each patch to patch combination? How were patches with vertically adjacent erosional and depositional areas handled (in panels with Discharges 1.5 and 2, particularly)?

   **Author's response- We did not measure every patch to patch combination but tried to capture the majority of the largest ones. We measured 48 lengths for each discharge. We did not match up vertically adjacent patches as sediment transport was assumed in a primarily downstream direction.**

2. I had to go back and reread multiple times to figure out how all of the different experimental runs that built the data into Figures 4 and 5. It might be easier if the "Flume and field data" section explicitly mentions how many experimental runs were completed at each discharge (or if they were collected sequentially, how many DEMs were collected for each experimental run)

**Author's response- Yes we agree and apologize for the lack of clarity. We have clarified this (see lines 273-276).**

Discussion

1. Line 306 – how can the method select an erroneous IMF? What physical basis exists to claim a selection is erroneous?

   **Author's response- We believe that the scale of the lower IMFs (1 and 2 especially) is too small to be meaningful i.e., on the scale of mm.**

2. Lines 307-320 – How can more than one distance be a suitable "characteristic path length"? It seems to me that the underlying evaluation of this method is the matching of erosion and deposition zones, which the paper claims is subjective. I think that subjectivity in methods is fine and often desirable, but that undercuts the stated need for an objective method.

   **Author's response- This was poorly worded and we have now tried to clarify throughout the manuscript that the decomposition is desirable to extract the underlying periodicity and not that all IMFs represent a characteristic path length but that perhaps a range can be estimated between IMF 4 and IMF 5. See sect. 5.1.4 using the IMFs.**

3. Line 321-323 - In my view, a weakness of this paper is the lack of clarity in explanation around the assumptions and simplifications inherent in the method. The signal processing method seems like it contains a lot of information across the IMFs that can be used to approximate or simplify topographically derived path lengths across the entirety of the DoD. Topographically-derived path lengths have their own assumptions, which are discussed in the discussion (throughput, compensating exchange), especially when compared with tracers (where the issue of whether or not the grain sizes of the tracer particles are representative of the topographically active sediment – See McDowell et al. 2021 for a discussion - as grain size has been observed to impact path length measurements – see Hassan and Bradley 2017). Like with tracers (McDowell et al., 2020), topographically estimated path lengths have been observed to vary with location along the channel (Calle et al., 2020, McDowell et al., 2021), complicating the implications of reducing these values to a simple number.

   **Author's response- We agree and have tried to clarify the assumptions and simplifications we make from the beginning and throughout the manuscript (see the extensive changes made throughout the manuscript).**

4. I think the paper would be strengthened if sediment budgets were determined for the laboratory experiments. This would allow for the use of the throughput index address the questions in 5.1 and strengthen the discussion about time-windowing.

   **Author's response- We computed the reach scale sediment budget for the flume experiments and report the volumes of erosion and deposition for the different time windows in Table A1. We preferred not to use the throughput index, as it cannot be easily computed in the field, as opposed to the proportion of active width. We did not want to further expand the paper analysis**

5. I like the discussion of the morphologically active width and Figure 10. It's a useful (and new, at least to me) warning for those who wish to apply the morphologic method.

    1. Change the X-axis label in Figure 10.

       **Author's response- Thanks, we are happy the figure is useful. We have changed this figure.**

---

## Author Comment (AC3)

**This comment is in response to anonymous reviewer 3 (RC3).**
**Thank you very much for taking the time to thoughtfully review our manuscript. We have worked to carefully consider all of your proposed suggestions and revisions. Below we will give responses to the specific questions.**

This manuscript presents a signal processing approach to estimate a characteristic sediment path length from DoDs, which in turn are used to estimate sediment transport. This work builds on a previous seminal work by Vericat et al. (2017), who anticipated the potential of DoDs combined with virtual sediment velocity (~path length) estimates to enrich the potential of the 'morphological approach' and quantify sediment fluxes in gravel-bed rivers. The topic is interesting, so the paper could potentially be of interest to the river science community and fits well within the scope of E*arth Surface Dynamics*.

The manuscript is very well written and organized. However, although the manuscript and the data presented seems in general to be good (at least to me), I found some crucial weaknesses. Consequently, I believe that the draft still needs some important work before its publication. Several reasons lead me to this assessment, which I will try to explain and justify in the following paragraphs. Nevertheless, I am confident that the authors will be able to adequately address these concerns and, after major reviews, prepare a manuscript that will be closer to acceptance.

**Hypothesis**: I have several concerns about the main hypothesis of this paper as stated by the authors. They explicitly write that their working hypothesis is that the "*sediment moves from one area of net deposition to an area of net erosion during the time period between DEM acquisition and that this represents a characteristic path length*" (lines 85-86). However, this statement needs some preliminary terminological clarification as to how the authors define and understand the term 'path length'. In my opinion, there are (at least) two different meanings for this term. On the one hand, it could be understood as the distance travelled by individual particles during a transport event; this is the meaning typically used in tracer studies. On the other hand, 'path length' could be understood as the average distance travelled by sediment during typical channel-forming flows. This is the meaning sometimes used in studies employing the morphological approach, which is typically equated with a characteristic morphological length scale of channel (meander length, bar-pool length, bar spacing, etc).

**Author's response- Thank you for pointing this out and we agree that this was not clear in the first version of the manuscript. To rectify this, we have extensively expanded the introduction to expand on the path length terminology. Further, we have reworded our objectives and clarified the underlying assumptions (see lines 120-128 and extensive changes to the introduction).**

Tracer studies have long shown that path lengths during transport episodes are not defined by a unique single value as assumed sometimes with the morphological approach, but by a distribution of values, whose exact shape (gamma, exponential, heavy vs thin-tailed…) is a widely discussed topic (with hundreds of pages of previous literature). Bearing all this in mind, it seems that by 'path length' the authors are somehow referring to the 'mean' or 'average' distance travelled by bedload particles during the sediment transport episodes studied. In this respect, the real hypothesis could be stated as: 'the distance between successive deposition and erosion patches, measured in a DoD, could be used as a proxy for the average travel distance (~path length) of the bed load'. If formulated in this way, the paper would provide continuity with previous research

highlighting some kind of morphological control on average bedload travel distances and would potentially enrich the debate on the topic.

**Author's response- Absolutely! We have expanded the introduction to discuss the idea of "path length" vs "step length" vs "characteristic path length" the latter of which is what we are trying to estimate. We have also updated the objectives to explain why we compare our estimates of a characteristic path length to the measured path length distributions in the field and how this gives support to the idea of morphological control on average bedload travel distances. Additionally, we bring this topic up in the discussion with regard to the tracer distributions.**

Apart from this, and although I appreciate the effort made by the authors in explicitly stating their hypothesis, in my opinion the manuscript does not clearly show how this hypothesis is tested. As a result, the reader cannot be sure or convinced if this hypothesis is adequately supported (or not) by the results. Then, I will encourage the authors to make more effort to link their methods to their hypothesis and to explicitly discuss the results in the light of the hypothesis.

**Author's response- We agree that the first version of the manuscript was not clear and have reworded the objectives and their underlying assumptions (see lines 120-128).**

**Introduction**: The review of previous literature presented in the introduction needs to be expanded, as I missed some recent works that are clearly relevant for this work. For instance, Bakker et al. (2019) (https://doi.org/10.1029/2018JF004811) presented a nice application of the morphological approach, based on DoD, where they presented a spatially distributed characterization of bedload in a braided river. I think this paper deserves (at least) a mention in the introduction. Then, there is the work done by Calle et al. (2020) (https://doi.org/10.1002/esp.4765), which is even more relevant because these authors present a workflow for estimating travel distances from DoD, which should be commented and discussed. What are the main novelties of the present manuscript compared to Calle et al. (2020)?

**Author's response- We agree that these are important studies to include and have expanded the introduction to include these articles and much more literature on applications of the morphological method. We have also tried to highlight the utility of our approach compared to the previous applications both in the introduction and discussion. (See extensive changes to the introduction and discussion).**

**Materials and Methods**: There are several aspects of the methods used in this paper that need clarification. In the absence of this information, it is difficult to assess the work of the authors.

For instance, in lines105-108, the authors comment that the DoD is divided "*into a series of equally sized bins*". However, there is no comment on the exact size chosen for these 'bins'. In addition, some sensitivity analysis of the effect of bin size on the final results would be welcomed.

**Author's response- We agree and have expanded the methods section to discuss bin size selection (see lines 171-178 and lines 600-605 in the discussion).**

Regarding their flume experiments, in line 146-147 they state that the flume was filled "*with sand characterized by a median diameter (D50) of 1 mm*". However, they do not specify whether this sediment is well- or poorly-sorted. Do you have any idea?

**Author's response- The sand was homogenously 1mm. We have clarified this (see lines 253-254).**

In addition, I have some concerns with the use of San Juan river data for the present work, as long as in this set of data, areas within the channel that were inundated in both the old and new DEMs were not used, thereby limiting the DoD analysis to areas above-water (MacQueen et al., 2021). Indeed, MacQueen et al. (2021) explicitly said: "*[DEM] were not used to calculate complete reach-scale sediment budgets due to the lack of in-channel topographic data and stage differences during each LiDAR survey affecting the relative portion of the river bed that was exposed*". This is a major limitation of the present manuscript. The authors should provide information on what proportion of the San Juan channel is represented by underwater areas and excluded from the DoDs. They also need to provide a better justification or discussion on how they believe this does not bias their results, and not simply reduce it to what they state in lines 190-192. They need to better justify lines 190-192 and explain why they believe that they can use San Juan data in a way that it was prevented by the original authors who collected this data (MacQueen et al., 2021).

**Author's response- We agree that this was a large oversight in the previous version of the manuscript and have now included information on the submerged area of the DoDs and have also explained why we believe this data is still relevant for our purposes (see lines 299-311).**

Results: In general, I think the presentation of results needs to be improved.

First, flume and field data are not treated in the same way, making it difficult to assess how their results do or not validate their hypothesis. I understand that this could be due to the different nature of the source data, but some effort should be done to homogenise the presentation of results. For example, the path lengths obtained from flume DoDs are validated against "*manually measured distances*" (line 231). Why is this not also done for the San Juan DoDs? By the way, there is nothing in the methods section (or anywhere else in the manuscript) about how this manual measurement of path lengths was done. This should be explained in at least a few lines, so readers could properly evaluate their work.

**Author's response- We agree that this was an issue as the original manuscript was written. We have now clarified that the San Juan River data is used to see how the characteristic path length of sediment transport might compare to a physical path length distribution. (See lines 135-137). We have expanded the section on the manual method (see lines 141-161).**

In addition, the presentation of results for the San Juan river is really poor (just one paragraph). To my knowledge, MacQueen et al. (2021) presented results from three or four tracer surveys in the San Juan River: 2016-2017, 2017-2018 and 2018-2019. Not much information is provided in section 4.2, so we don't know if all the numbers presented in this section refer to the average of all these tracer surveys, to just one survey… However, in figure 7 it seems that the authors are using just one survey. Why is this?

**Author's response- We agree that the previous result section for the San Juan River data was very sparse and has now been expanded (see Sect. 4.2) however because this was more of a qualitative comparison, we use the discussion to explore the idea of how a characteristic path**

length compares to the distributions (see lines 455-477).  We apologize for the lack of clarity in the previous version of the manuscript and have now clarified that there were only DoDs corresponding to the tracer deployment for one year, 2018-2019. The other DoD spanned multiple tracer deployments and therefore we not desirable for our purposes (see lines 248-251).

Discussion: Again, I have some concerns about how the authors discussed their results.

In lines 284-291, the authors attempt to discuss the fact that their path length estimates do not change with flow discharge. This seems counterintuitive, as we would expect average travel distances to increase with flow intensity, as many decades of previous tracer research have already shown. The authors managed to put forward two possible explanations, but I am not entirely satisfied with how they present and discuss them.

The first proposed explanation is that at higher discharges the characteristic path length is greater than the average distance between erosion and deposition patches (which I think is probably the correct explanation). But in fact this is not an explanation, but a bias in their method, which results from their assumption that the sediment simply moves from a deposition patch to the closer erosion patch downstream. However, sediment can travel (and often does) further than a morphological unit in gravel-bed rivers. Indeed, there is some field support for this in previous tracer data. Vázquez-Tarrío et al. (2019) conducted a meta-analyse of previously published tracer data, in which they compare the mean travel distances of sediment to the morphological length scale of the channel. They observed that at high discharges, mean travel distances of tracers can be larger than one morphological unit. The same was observed by Liebault et al. (2012) in a wandering river. In this respect, some discussion could be done on how to exploit tracer data to better constrain how the average number of morphological units travelled by bedload particles increases with discharge. This could be a clue on how to improve in the future their method.

**Author's response- Yes, these are great points! We have expanded the discussion to engage with these ideas (see sect 5.1.1 Flow effects).**

In this respect, the approach presented by Calle et al. (2020) seems to be more appropriate to deal with these situations than the one presented in the present manuscript. Calle et al. (2020) attempt to account for this increase in travel distances with flow discharge by using the balances between the volumes of erosion and deposition. Consequently, this previous work of Calle et al. (2020) should be discussed here. However, it is not mentioned in the whole manuscript.

**Author's response- We recognize the importance of including the work of Calle et al. (2020). and have included references to their study. We have also tried to emphasize that our study is focused on a different objective (the characteristic path length as represented by the periodicity of erosion and deposition).**

Another issue follows from this fact that their method is insensitive to flow discharge. Indeed, the average distance between erosion and deposition patches should scale with channel dimensions, i.e., in principle, we could expect larger distances between erosion and deposition sites in larger channels and smaller distances in smaller channels. Then, if the method is insensitive to flow discharge, we might ask what the advantage of such a sophisticated approach is, compared to the more straightforward procedure of simply considering the channel width (as Beechie, 2001) or the channel width times 5-7 (typical values for average pool spacing in gravel bed rivers) as the

characteristic path length. Some discussion of this should then be included. In doing so, the authors will potentially provide interesting information on three questions. First, is their signal processing method more efficient or not than simply using the channel dimensions as a proxy for path length? This is an interesting result in itself, as it could help to simplify future works. Second, is there a consistent relationship between the distance of erosion-sediment patches observed in the DoDs and the channel dimensions (width, pool-pool spacing, etc)? Finally, third, is this scaling consistent between flume and field data?

**Author's response- Thank you for raising these questions! We have conducted additional analysis and included the results which we believe strengthen the manuscript. Additionally, we have expanded the discussion to include these concepts (see sects. 5.1.1, 5.1.3, and appendix figures A6 and A7).**

Finally, Redolfi (in Vericat et al., 2017) presents an alternative for estimating the path length, which consists in using the length scale of the long axis of erosion patches in DoDs as a proxy for the average path length of sediment. Some discussion should be done in this regard and authors should explain why they believe their approach is (or not) more adequate than the one proposed by Redolfi.

**Author's response- We agree that this was a major oversight of the original version of the manuscript and thank you for brining it to our attention. We have now included the work done by Redolfi (2014) and Vericat et al., (2017) in the introduction (see lines 103-109) and have made extensive changes to the manuscript in hopes of explaining why we believe that our method can contribute and build on the work done by Redolfi, Vericat et al., and others by providing a new way to think about the periodicity of erosion and deposition.**

---

## Editor Decision (ED1)

I feel that the authors have carefully considered all the points I made in my comments. I am very grateful for the cooperative attitude of the authors and therefore recommend publication of the paper, subject to some minor changes.

I must say that I appreciate the work done by the authors in reviewing their manuscript in the light of all the reviewers' comments; not only mine, but also those of the other two colleagues who also reviewed the manuscript.

However, my main remaining concern with the manuscript relates to the way in which the authors justify the importance of their work in the introduction. In order to emphasise the interest and importance of their approach, the authors somehow underestimate the value of tracer research. For instance, in lines 79-82 authors said that "*is unclear if the mean path length as measured by tracers is the best representation of a 'characteristic' path length to estimate bedload transport*" and they go on to say that the morphological approach could be used to infer a characteristic path length from changes in morphology as an alternative to tracers (lines 91-93). In my opinion the two methods are not antagonistic but complementary. In fact, tracers are the only method that provides real information about particle displacements. Inferences made by the morphological approach are surrogate estimates based on assumptions and hypotheses that may be questionable in some circumstances. Although I recognise the value of the morphological approach, that concern should not be forgotten and therefore I believe that both methods are complementary. Although my comment does not preclude my recommendation for acceptance of this manuscript, I would be grateful if the authors would consider it and perhaps rephrase some parts of the introduction.

Apart from that, I have a few minor comments. Before I list them, I would like to congratulate the authors on their work.

**Minor comments**

- Line 67: I think a comma is missing between "(Hoey, 1992; McLean and Church, 1999)" and "measurement uncertainty".

- Line 88: I think eq.1 deserves some reference. Maybe Ashmore and Church (1998) or Church (2006) should be cited.

- Lines 49-59 and eq. 2: The virtual velocity approach not only needs an estimation of path lengths, but also some estimation of the average active layer thickness ($d_s$ in eq. 2). Estimation of this thickness is also tricky. Tracers and scour chains have been used to measure it in the field, whereas in the morphological approach it is estimated using DoDs. I think some mention and references would be welcomed. For instance, Church and Haschenburger (https://doi.org/10.1002/2016WR019675), Leduc and Ashmore (https://doi.org/10.1002/2017WR022438), Vázquez Tarrio et al. 2020 (https://doi.org/10.1002/esp.5027) , etc.

- Line 80-82: This long sentence is difficult to understand, at least for non-native English speakers. Please, consider modifying it.

- Line 82-84: Same remark as above.

- Lines 84-90: Here authors state that "*at formative discharges, particle path length distributions often exhibit primary or secondary modes corresponding to the location of bars, where deposition occurs*", so "*the characteristic path length, i.e., the most representative and sound value to be used in sediment transport estimations, might be*

*better described by these primary or secondary modes in channels with bar morphology at channel forming flows*".

I agree with this comment. But the authors seem to use these statements as an argument against the virtual velocity approach based on mean travel distances derived from tracer data (eq. 2) and in favour of the morphological approach (eq. 3). However, in my opinion, both methods suffer from the same problem. The morphological approach used by the authors estimates the characteristic path length as the distance between an erosion patch and a subsequent deposition patch. This neglects the possibility that particles can travel further from a single morphological unit, so it also implies that secondary modes of particle displacement are not considered. This relates to my previous comment and I sincerely believe that the authors should consider rephrasing some sentences.

- Lines 141-142: This brings us back to my same recurring concern. To me, this hypothesis contradicts your statements in lines 84-90. If there are secondary (and more) modes of particle displacement that bias tracer studies... how can you be sure that these secondary modes do not also bias your method? As long as this method assumes that sediment does not travel further than the distance between one erosion-deposition couplet...this seems to contradict your literature review in the introduction, doesn't it?

- Lines 305-306: This may deserve further justification. Firstly, what percentage of the area is covered by the underwater channel? We need this information to be sure that the underwater channel does not represent a large percentage of the surveyed area in the San Juan River.

  Then I think your statement (that the underwater channel may not distort the patterns of erosion/deposition) would only be true if the underwater channel only acts as a sediment conveyor belt. However, if this is not the case, even if it is only a small part of the area studied, then the overall patterns of erosion/deposition in the underwater channel could well confound the path length estimates. For example, we could imagine that the underwater channel could be preferentially eroded at channel confluences/flow convergence areas, or show persistent deposition at channel diffluences or secondary flow areas.

- Lines 425-427: Again, this seems for me somehow contradictory with lines 84-90 and the idea that there are secondary and larger modes of particle displacements.

- Line 455-456: Be careful with this. What is the tracer recovery? Where are the missing tracers? Recovery rates reported by McQueen et al. (2019) in San Juan river are relatively high: around 70-75%. However, if the missing 25-30% were preferentially tracer frontrunners, then this statement could be biased.

- Line 474-477: This is a nice support to the idea that path lengths may probably increase with discharge.

- Section of Confinement: I personally like the originality of this addition to the discussion.

- Line 509-521: I appreciate that the authors have considered my previous suggestions.

---

## Author Response (AR2)

Response to referee report (version 2)-

**We would like to thank you very much for taking more time to review our article and greatly value your contribution. We believe that we have addressed each of your concerns below.**

However, my main remaining concern with the manuscript relates to the way in which the authors justify the importance of their work in the introduction. In order to emphasise the interest and importance of their approach, the authors somehow underestimate the value of tracer research. For instance, in lines 79-82 authors said that "is unclear if the mean path length as measured by tracers is the best representation of a 'characteristic' path length to estimate bedload transport" and they go on to say that the morphological approach could be used to infer a characteristic path length from changes in morphology as an alternative to tracers (lines 91-93). In my opinion the two methods are not antagonistic but complementary. In fact, tracers are the only method that provides real information about particle displacements. Inferences made by the morphological approach are surrogate estimates based on assumptions and hypotheses that may be questionable in some circumstances. Although I recognise the value of the morphological approach, that concern should not be forgotten and therefore I believe that both methods are complementary.

**Author's response- We agree that the two methods should be complementary, especially now in the development of this new method. We meant to say that perhaps the mode, or secondary mode, of the path length distribution is more meaningful to estimate sediment transport. So it is not a criticism of tracer studies, just that perhaps the *mean* is not always the ideal value. The identification of these secondary modes which correspond to the location of bars would be impossible without these tracer studies so we certainly do not want to downplay their importance. We have now re-worded these lines for clarity. (see lines 82-86)**

**Additionally to your point, we think that perhaps by using tracers and our method in situations with varying discharge, we can identify a threshold for when the majority of particles are traveling further than one morphological unit, and multiples of the path length estimated by the method are needed. For example the 2 l/s discharge (see discussion sect 5.1.1).**

Minor comments  -

 Line 67: I think a comma is missing between "(Hoey, 1992; McLean and Church, 1999)" and "measurement uncertainty".  (**referring to the tracked changes doc)

**Author's response- This has now been fixed (see line 23).**

- Line 88: I think eq.1 deserves some reference. Maybe Ashmore and Church (1998) or Church (2006) should be cited. (**referring to the tracked changes doc)

**Author's response- We agree and have added these references (see line 44).**

 - Lines 49-59 and eq. 2: The virtual velocity approach not only needs an estimation of path lengths, but also some estimation of the average active layer thickness (ds in eq. 2). Estimation of this thickness is also tricky. Tracers and scour chains have been used to measure it in the field, whereas in the morphological approach it is estimated using DoDs. I think some mention and references would be welcomed. For instance, Church and Haschenburger

(https://doi.org/10.1002/2016WR019675), Leduc and Ashmore (https://doi.org/10.1002/2017WR022438), Vázquez Tarrio et al. 2020 (https://doi.org/10.1002/esp.5027) , etc.

**Author's response- We agree this was an oversight and now have included a line to describe this term with citations. (see lines 55-56).**

 - Line 80-82: This long sentence is difficult to understand, at least for non-native English speakers. Please, consider modifying it. (\*\*referring to the tracked changes doc)

**Author's response- Thank you for pointing this out, we agree and have re-worded this sentence (see lines 31-33).**

- Line 82-84: Same remark as above. (\*\*referring to the tracked changes doc)

**Author's response- See lines (34-37)**

- Lines 84-90: Here authors state that "at formative discharges, particle path length distributions often exhibit primary or secondary modes corresponding to the location of bars, where deposition occurs", so "the characteristic path length, i.e., the most representative and sound value to be used in sediment transport estimations, might be better described by these primary or secondary modes in channels with bar morphology at channel forming flows".   I agree with this comment. But the authors seem to use these statements as an argument against the virtual velocity approach based on mean travel distances derived from tracer data (eq. 2) and in favour of the morphological approach (eq. 3). However, in my opinion, both methods suffer from the same problem. The morphological approach used by the authors estimates the characteristic path length as the distance between an erosion patch and a subsequent deposition patch. This neglects the possibility that particles can travel further from a single morphological unit, so it also implies that secondary modes of particle displacement are not considered. This relates to my previous comment and I sincerely believe that the authors should consider rephrasing some sentences.

**Author's response- We think this comes from the lack of clarity in the introduction. Hopefully, by changing lines (82-86) we have shown that it was not meant as an argument against the virtual velocity approach using tracers, just that the mean value is perhaps not the most representative. By rewording the introduction, we have tried to emphasize that tracer studies are fundamental in creating the relationship between path length and morphology. We have tried to clarify this in the introduction (see lines 82-87)**

Lines 141-142:  This brings us back to my same recurring concern. To me, this hypothesis contradicts your statements in lines 84-90. If there are secondary (and more) modes of particle displacement that bias tracer studies... how can you be sure that these secondary modes do not also bias your method? As long as this method assumes that sediment does not travel further than the distance between one erosion-deposition couplet...this seems to contradict your literature review in the introduction, doesn't it?

**Author's response- Thank you for bringing this to our attention. We were not clear in the introduction. It was not meant to be argued that these secondary modes bias tracer studies, but that these tracer studies have shown the importance of bars as depositional sites as evidenced by the secondary modes. Using tracers to estimate the virtual velocity is absolutely acceptable in our opinion and the mode or secondary mode could be considered as the path length (see lines 82-87). We have also seen that in the McQueen data, our method does pick up on these secondary modes, so it is not necessarily a bias in our method. What is a major fault**

in our method is the inability to pick up on multiples of the bar spacing. Tracers could and should help us in the future define the conditions under which sediment is traveling multiple morphological units and potentially define a threshold for using multiples of the characteristic path length to estimate sediment transport (see discussion lines 471-480).

- Lines 305-306: This may deserve further justification. Firstly, what percentage of the area is covered by the underwater channel? We need this information to be sure that the underwater channel does not represent a large percentage of the surveyed area in the San Juan River.

**Author's response- We agree that the percentage of submerged channel should be included and have updated this section to include that information and also tried to explain that this could bias the results but given the limited available datasets containing quality DoDs and RFID tracer data, we think it is an appropriate first application (see lines 305-315). We are currently working on a study with DoDs that includes the submerged area and hope to have more chances to apply the method in the future.**

- Lines 425-427: Again, this seems for me somehow contradictory with lines 84-90 and the idea that there are secondary and larger modes of particle displacements.

**Author's response- We hope that clarifying what we meant in the introduction will address this concern (lines 82-86). What we mean here is that the secondary modes from Pyrce and Ashmore's study were often the first bar downstream of the insertion location. So the first mode was generally the tracers that moved very little from the point of insertion and the secondary mode was the first depositional unit downstream. This is what we mean by the next depositional unit downstream.**

- Line 455-456: Be careful with this. What is the tracer recovery? Where are the missing tracers? Recovery rates reported by McQueen et al. (2019) in San Juan river are relatively high: around 70-75%. However, if the missing 25-30% were preferentially tracer frontrunners, then this statement could be biased.

**Author's response- Good point! We have included this caveat. (see lines 462-465).**

- Line 474-477: This is a nice support to the idea that path lengths may probably increase with discharge.

**Author's response- Yes, we agree and hope to conduct further studies to verify this hypothesis.**

- Section of Confinement: I personally like the originality of this addition to the discussion.

**Author's response- Thank you!**

- Line 509-521: I appreciate that the authors have considered my previous suggestions.

**Author's response- Thank you for your valuable suggestions. They greatly improved the quality and completeness of the article.**